# The liver-enriched lnc-LFAR1 promotes liver fibrosis by activating TGFβ and Notch pathways

Kun Zhang[1], Xiaohui Han[1], Zhen Zhang[1], Lina Zheng[1], Zhimei Hu[1], Qingbin Yao[1], Hongmei Cui[1], Guiming Shu[2], Maojie Si[2], Chan Li[2], Zhemin Shi[1], Ting Chen[1], Yawei Han[1], Yanan Chang[1], Zhi Yao[3], Tao Han[2,4,5] & Wei Hong[1]

Long noncoding RNAs (lncRNAs) play important roles in various biological processes such as proliferation, cell death and differentiation. Here, we show that a liver-enriched lncRNA, named liver fibrosis-associated lncRNA1 (lnc-LFAR1), promotes liver fibrosis. We demonstrate that lnc-LFAR1 silencing impairs hepatic stellate cells (HSCs) activation, reduces TGFβ-induced hepatocytes apoptosis in vitro and attenuates both $CCl_4$- and bile duct ligation-induced liver fibrosis in mice. Lnc-LFAR1 promotes the binding of Smad2/3 to TGFβR1 and its phosphorylation in the cytoplasm. Lnc-LFAR1 binds directly to Smad2/3 and promotes transcription of TGFβ, Smad2, Smad3, Notch2 and Notch3 which, in turn, results in TGFβ and Notch pathway activation. We show that the TGFβ1/Smad2/3/lnc-LFAR1 pathway provides a positive feedback loop to increase Smad2/3 response and a novel link connecting TGFβ with Notch pathway. Our work identifies a liver-enriched lncRNA that regulates liver fibrogenesis and suggests it as a potential target for fibrosis treatment.

[1] Department of Histology and Embryology, School of Basic Medical Sciences, Tianjin Medical University, Tianjin 300070, China. [2] The Third Central Clinical College of Tianjin Medical University, Tianjin Third Central Hospital, Tianjin 300170, China. [3] Department of Immunology, School of Basic Medical Sciences, Tianjin Medical University, Tianjin 300070, China. [4] Department of Hepatology, Tianjin Third Central Hospital, Tianjin 300170, China. [5] Tianjin Key Laboratory of Artificial Cells, Tianjin Third Central Hospital, Tianjin 300170, China. Kun Zhang and Xiaohui Han contributed equally to this work. Correspondence and requests for materials should be addressed to T.H. (email: hantaomd@126.com) or to W.H. (email: hongwei@tijmu.edu.cn)

L iver fibrosis is characterized by the pathological accumulation of extracellular matrix (ECM) components in the liver, which eventually leads to hepatic dysfunction. This process is caused by the persistent liver damage and wound-healing reaction induced by various insults including alcohol abuse, hepatitis virus and other etiologies and can progress to cirrhosis[1, 2]. A better understanding of the molecular mechanisms controlling the fibrotic response is needed to develop novel clinical strategies. It is generally accepted that activated hepatic stellate cells (HSCs) is the most principal cellular players promoting synthesis and deposition of ECM proteins in response

to accumulated levels of inflammatory signals derived from damaged parenchymal cells. In healthy liver, HSCs remain in a quiescent state[3], but following continued liver injury, quiescent HSCs trans-differentiate into myofibroblast-like cells that are characterized by the expression of α-SMA, and enhanced production of ECM. Activated HSCs respond to and secrete a variety of pro-fibrogenic cytokines including CTGF, TIMPs and TGFβ, which are the potent cytokines resulting in liver fibrosis[3]. Despite the fact that HSCs play a pivotal role in liver fibrosis, hepatocyte (HC) is the dominant cell type residing in the liver and HCs apoptosis and impaired HCs proliferation also have

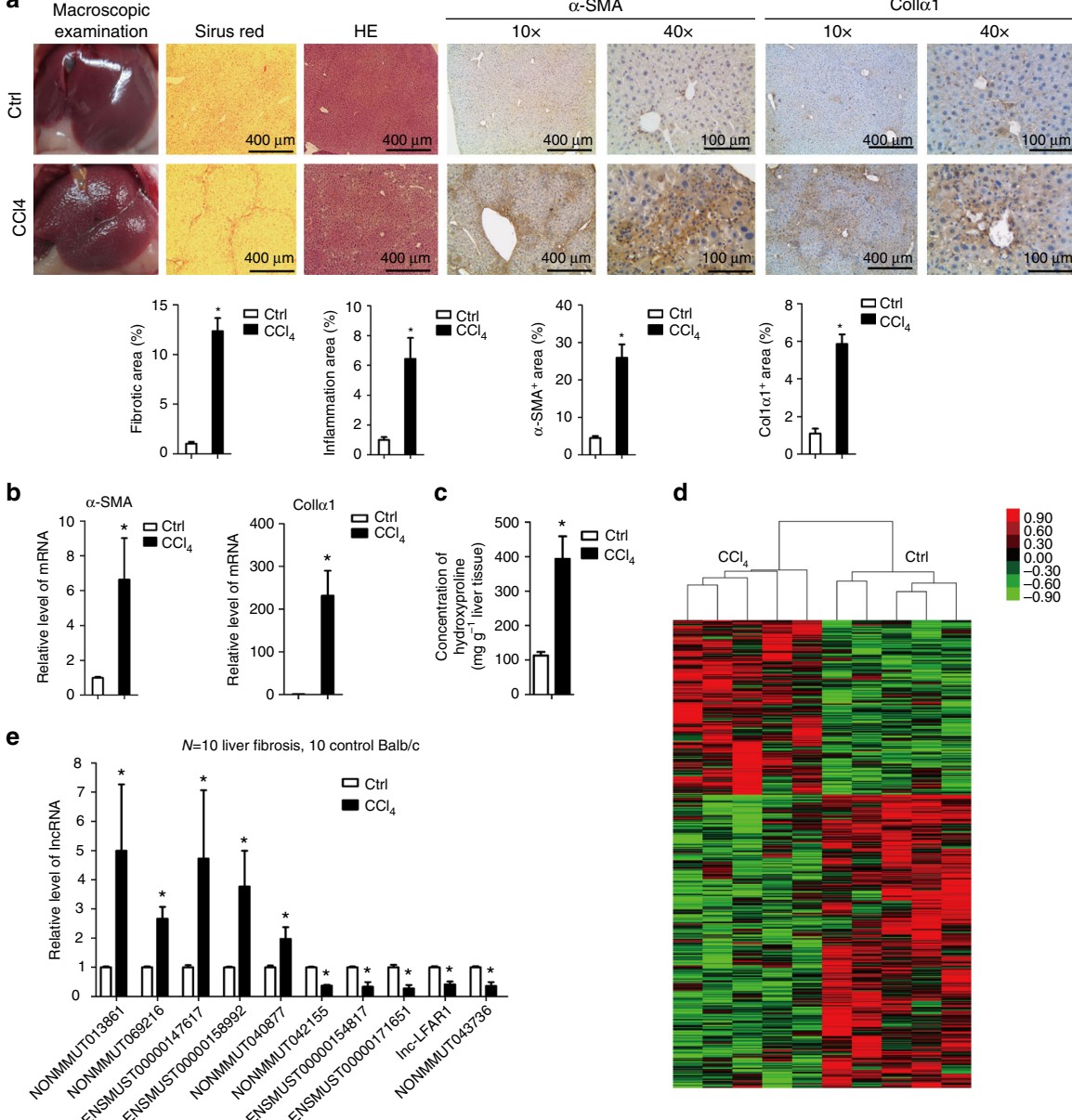

**Fig. 1** Expression profiles of lncRNAs on induction of liver fibrosis. Mice were injected with either olive oil or CCl4 to induce liver fibrosis for 6 weeks and were killed 2 days after the last injection. **a** Liver fibrosis was confirmed by macroscopic examination, H&E staining, Sirius red staining and IHC for α-SMA and collagen1 (n = 5 per group). Scale bars, 400 μm for H&E staining, Sirius red staining and IHC (objective, ×10); 100 μm for IHC (objective, ×40). Below, five images of each liver and five livers from different mice were quantified for each group. **b** qRT-PCR analysis of α-SMA and Col1α1 in the liver tissues upon injury (n = 5 per group). **c** Quantification of hepatic hydroxyproline content. The data are expressed as hydroxyproline (μg) per liver wet weight (g) (n = 5 per group). **d** Microarray analysis for lncRNA was performed with RNA extracted from livers of CCl4-treated (n = 5) and oil-treated (n = 5) Balb/c-mice (6 weeks' treatment). Hierarchical cluster analysis of significantly differentially expressed lncRNAs: bright green, under-expression; gray, no change; bright red, over-expression. **e** Differential expression of ten representative lncRNAs was validated in fibrotic and normal liver tissues by qRT-PCR in Balb/c mice (n = 10 per group). Data are presented as means ± s.e.m. P values were analyzed by Student's t-test. *P < 0.05

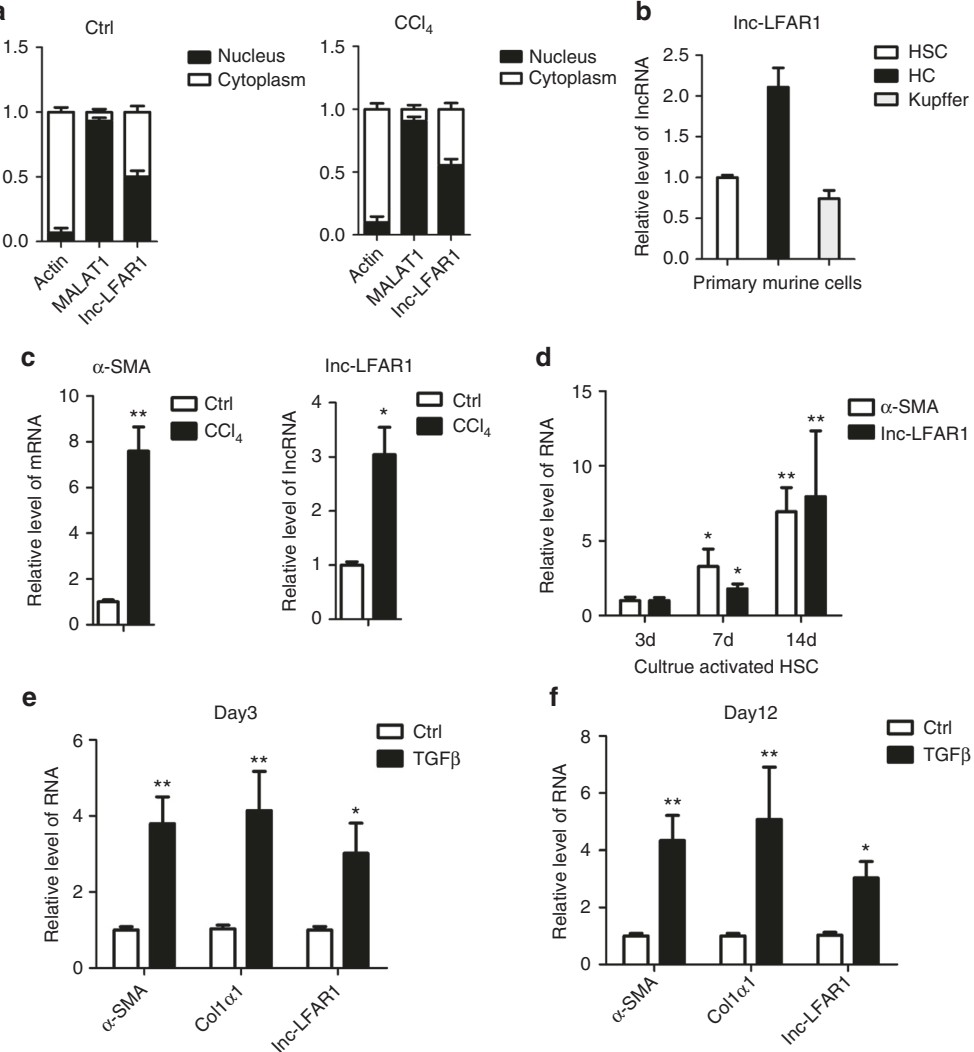

**Fig. 2** Lnc-LFAR1 is over-expressed in HSCs during liver fibrogenesis. **a** RNA was extracted from the nuclei or cytoplasm of primary HSCs isolated from normal and fibrotic livers. A total of 1 μg of RNA was used for the qRT-PCR analysis of lnc-LFAR1, lnc-MALAT1 (nuclear retained), and Actin mRNAs (cytoplasm retained). **b** HCs, HSCs, and Kupffer cells were isolated from livers of normal male Balb/c mice, and relative lnc-LFAR1 expression compared to HSCs was analyzed by qRT-PCR. **c** Primary HSCs were isolated from livers of Balb/c mice treated for 6 weeks with CCl₄ or oil, and the expression of *lnc-LFAR1* and *α-SMA* was determined by qRT-PCR. **d** *α-SMA* and *lnc-LFAR1* levels were measured by qRT-PCR in HSCs after culture-induced activation. **e**, **f** Primary HSCs cultured at day 3 **e** and day 12 **f** were stimulated with TGFβ for 24 h and the expression of *lnc-LFAR1* was determined by qRT-PCR. The number of biological replicates for each experiment was *n* ⩾ 3. Data are presented as means ± s.e.m. *P* values were analyzed by Student's *t*-test. *$P < 0.05$, **$P < 0.01$

been commonly recognized as critical initiators of fibrosis by activating HSCs in persistent liver injury[1]. Thus, the inactivation of HSCs and inhibition of HCs apoptosis have been currently accepted for the resolution of liver fibrosis.

The number of human protein-coding genes is less than 2% of the whole genome sequence, whereas the vast majority of transcripts consist of the noncoding RNAs, among which are long noncoding RNAs (lncRNAs) that are transcribed mainly by RNA polymerase II, 5'-capped and polyadenylated like most mRNAs, yet this class of transcripts has limited coding potential[4]. Despite their poor conservation and low levels of expression compared with protein-coding genes, lncRNAs are often regulated by transcription factors and are expressed in a cell- or tissue-specific manner[5, 6]. Recent reports have demonstrated that lncRNAs participate in modulating biological processes through regulating gene expression by a variety of mechanisms according to the cellular location[7]. With multiple and diverse targets, lncRNAs are involved in numerous biological functions and pathological

processes, including development, proliferation, apoptosis, survival, differentiation and carcinogenesis[8–13]. The specific contribution of selected lncRNAs in hepatic disease progression has been described. Recent studies reported the process of HSC trans-differentiation is governed by differential lncRNAs[14–17]. For instance, forced expression of GAS5 suppresses the activation of primary HSCs in vitro and alleviates the accumulation of collagen in fibrotic liver tissues in vivo by increasing p27 expression as a ceRNA for microRNA-222[15]. Moreover, it has been reported that over-expression of MEG3 could activate p53, subsequently leading to caspase-3-dependent apoptosis in TGFβ-treated LX-2 cells[16]. Additional study reported that H19 is hardly detectable in adult liver but is markedly increased in fibrotic/cirrhotic human and mouse liver[17]. Although the field is developing, studies to date have lacked accurate lncRNA profiling of the fibrotic liver tissue. Additionally, no studies have identified any lncRNAs with global effect on pro-fibrotic signaling in the liver, which could be more efficient than targeting a single gene.

In this study, we determine the lncRNA expression profile in the livers of fibrotic mice and normal mice by lncRNA microarrays and real-time PCR. Through a detailed analysis of the expression of lncRNAs in various tissues, we discover a liver-enriched lncRNA-LFAR1 (liver fibrosis-associated lncRNA1) and define its expression profile and function. We show that, despite downregulated lnc-LFAR1 level in the whole liver lncRNA extracted from the fibrotic mice, lnc-LFAR1 is specifically upregulated in HSCs during fibrogenesis. This upregulation is mediated by TGFβ, and promotes HSCs activation and TGFβ-induced HCs apoptosis. Mechanistically, we demonstrate that lnc-LFAR1 promotes the association of Smad2/3 with TGFβR1 which subsequently phosphorylates Smad2/3 in the cytoplasm. Moreover, we validate lnc-LFAR1 interacts with the transcriptional factor Smad2/3 by RIP assay and our data suggest that knockdown of lnc-LFAR1 dramatically inactivates fibrotic TGFβ/Smad and Notch pathways in both HSCs and HCs and thereby inhibiting $CCl_4$- and bile duct ligation (BDL)-induced mouse liver fibrosis in vivo. This study may provide a mechanism and potential therapeutic approach for treating hepatic fibrosis.

## Results

**LncRNAs expression profile in liver fibrosis mouse model**. In a systematic approach to identify lncRNAs involved in liver fibrosis, we applied the well-established model of $CCl_4$ treatment for hepatic fibrogenesis in mice. Firstly, liver fibrosis was induced in Balb/c mice by injecting $CCl_4$ for 6 weeks. As shown in Fig. 1a and Supplementary Fig. 1a, macroscopic examination, hematoxylin-eosin (H&E) staining, Sirius red staining, immunofluorescent assay for collagen1 and immunohistochemistry (IHC) for α-SMA and collagen1 confirmed the occurrence of liver fibrosis. The fibrotic livers exhibited significant upregulation of the mRNAs of *α-SMA* and *Col1α1* compared with the control livers (Fig. 1b). Moreover, the concentration of hydroxyproline in mouse livers also increased from 104 to 397 μg g$^{-1}$ after $CCl_4$ injection (Fig. 1c). We then applied microarray analysis to compare lncRNA and mRNA expression levels between fibrotic livers and normal livers, and found 266 lncRNAs and 1007 mRNAs were upregulated, 447 lncRNAs and 519 mRNAs were downregulated in the fibrotic livers (Fig. 1d and Supplementary Fig. 1b). To validate the findings of microarray analysis, we selected 10 lncRNAs according to the fold change and the lncRNA-mRNA co-expression network, 10 mRNAs that are related to liver fibrosis from the results of microarray, and analyzed their expression in another 10 pairs of fibrotic livers and normal livers from Balb/c and C57 mice, respectively (Fig. 1e and Supplementary Fig. 1c–e). The results confirmed that NONMMUT013861, NONMMUT069216, ENSMUST00000147617, ENSMUST00000158992 and NONMMUT040877 were over-expressed in fibrotic liver, whereas the expression of NONMMUT042155, ENSMUST00000154817, ENSMUST00000171651, NONMMUT045304 (lnc-LFAR1) and NONMMUT043736 were decreased. Thus, by applying a systematic array approach, we identify subsets of lncRNAs that are differentially regulated during $CCl_4$-induced liver fibrosis.

**Identification of a liver-enriched Lnc-LFAR1**. To identify lncRNAs that are potentially involved in liver fibrosis, we firstly searched for lncRNAs that are enriched in liver. Through a detailed analysis of the expression of the identified subsets of lncRNAs in various tissues of normal and fibrotic mice, we identified a liver-enriched lncRNA, and we named it liver fibrosis-associated lncRNA1 (lnc-LFAR1; Supplementary Fig. 2a). Cell fractionation followed by quantitative polymerase chain reaction with reverse transcription (qRT-PCR) showed that lnc-LFAR1 located both in the cytoplasm and the nucleus of primary HSCs from normal and fibrotic livers (Fig. 2a). Similar results were obtained from primary HCs and AML12 cells (Supplementary Fig. 2b). With the use of 5'- and 3'-rapid amplification of cDNA ends (RACE), lnc-LFAR1 was found to be a 734-nucleotide transcript comprising only one exon, consistent with our microarray analysis (Supplementary Fig. 2c). Because lnc-LFAR1 is a novel transcript, we determined whether it represents a protein-coding gene. Although lnc-LFAR1 harbors short open reading frame (ORF) of 55aa (Supplementary Fig. 3a), lnc-LFAR1 lacks Kozak sequence, which is important for translation initiation. The full-length transcript has no protein-coding potential according to the coding potential calculator (CPC). In addition, we cloned the predicted ORF of lnc-LFAR1 into pcDNA3.1 (+) vector, with a C-terminal enhanced green fluorescent protein (EGFP) tag (Supplementary Fig. 3b). GAPDH ORF was used as positive control, and lnc-MALAT1 ORF was used as negative control. We transiently transfected the EGFP-tagged expression vectors into AML12 cells, and immunofluorescence showed that EGFP is hardly detected in lnc-LFAR1 or lnc-MALAT1 group. However, it is easily detectable in GAPDH group (Supplementary Fig. 3c). On the basis of the informatics analysis and experimental evidences, we conclude that lnc-LFAR1 is a lncRNA.

To assess the function of lnc-LFAR1 in liver fibrosis, we firstly evaluated lnc-LFAR1 expression in different cell types in liver. The highest expression level of lnc-LFAR1 was found in HCs followed by HSCs and Kupffer cells (Fig. 2b). Lnc-LFAR1 expression was next compared in primary HSCs and HCs isolated from normal and fibrotic mice respectively. Expectedly, lnc-LFAR1 expression was downregulated in primary HCs isolated from fibrotic mice (Supplementary Fig. 4a), whereas increased in primary HSCs of fibrotic mice (Fig. 2c). Furthermore, primary HSCs at days 3, 7 and 14 were also collected to evaluate lnc-LFAR1 expression during HSC activation. As shown in Fig. 2d, the expression of α-SMA was significantly increased at day 14, consistent with previous report that HSCs cultured at day 14 are deemed as the fully activated[3]. Activated HSCs exhibits significant upregulation of the expression of lnc-LFAR1 at day 14, compared with day 3. Finally, we tested whether TGFβ might regulate the expression of lnc-LFAR1 in HSCs. In line with the upregulation of lnc-LFAR1 in HSCs during hepatic fibrogenesis, stimulation of primary HSCs with recombinant TGFβ resulted in a significant increased level of lnc-LFAR1 in both days 3 and 12, correlating with an increase of α-SMA and Col1α1 expression in these cells (Fig. 2e, f). In addition, we found that TGFβ also increases lnc-LFAR1 expression in primary HCs and AML12 cells (Supplementary Fig. 4b, c), which is inconsistent with the previous results (Supplementary Fig. 4a). Therefore, we investigated the expression of lnc-LFAR1 in liver tissues of mice treated with $CCl_4$ for 2, 4, 6, 8 or 10 weeks. Interestingly, the level of lnc-LFAR1 is decreased drastically at 2 weeks after $CCl_4$ injection, while gradually increased with persists $CCl_4$ injection (Supplementary Fig. 4d), and these phenomena were also observed in mice that underwent BDL for 3, 14 or 21 days and primary HCs isolated from mice treated with $CCl_4$ for 2, 4, 6, 8 or 10 weeks (Supplementary Fig. 4e, f), suggesting that lnc-LFAR1 may differentially express during the different stages of liver fibrosis. Taken together, lnc-LFAR1 is differently expressed in various cells and stages during fibrogenesis.

**Lnc-LFAR1 regulates the expression of ECM genes in HSCs**. Activation of HSCs represents a key feature of liver fibrosis and is characterized by specific gene expression patterns such as high

collagen expression[18]. To further evaluate the functional role of lnc-LFAR1 in liver fibrosis, we have knocked down lnc-LFAR1 with two separated lnc-LFAR1-shRNAs in primary HSCs and isolated RNA for RNA-seq. The data revealed that 1195 mRNAs were upregulated and 1424 mRNAs were downregulated in the primary HSCs infected with lncRNA-shRNA-1. Among these, 2023 out of 2619 genes, including α-SMA, Col1α1, Col1α2, Col3α1, Col4α5, TGFβR1, MMPs and TIMPs, were in common with the set of genes deregulated in the primary HSCs infected with lncRNA-shRNA-3 (Fig. 3a–c). The GO analysis revealed

that lnc-LFAR1 silencing affects a list of genes associated with ECM and the KEGG pathway analysis demonstrated ECM-receptor interaction pathway (Figs. 3d, e), suggesting that lnc-LFAR1 regulates the expression of ECM genes in HSCs. In addition, we also used lentivirus vector of the two separated lnc-LFAR1-shRNAs to knockdown its expression in primary HSCs at days 2 and 12. Recombinant TGFβ was then added to HSCs 72 h after infection with lnc-LFAR1-shRNAs or shRNA-control virus, and total RNA was extracted for detection of the expression of fibrosis-related genes. We found that cells

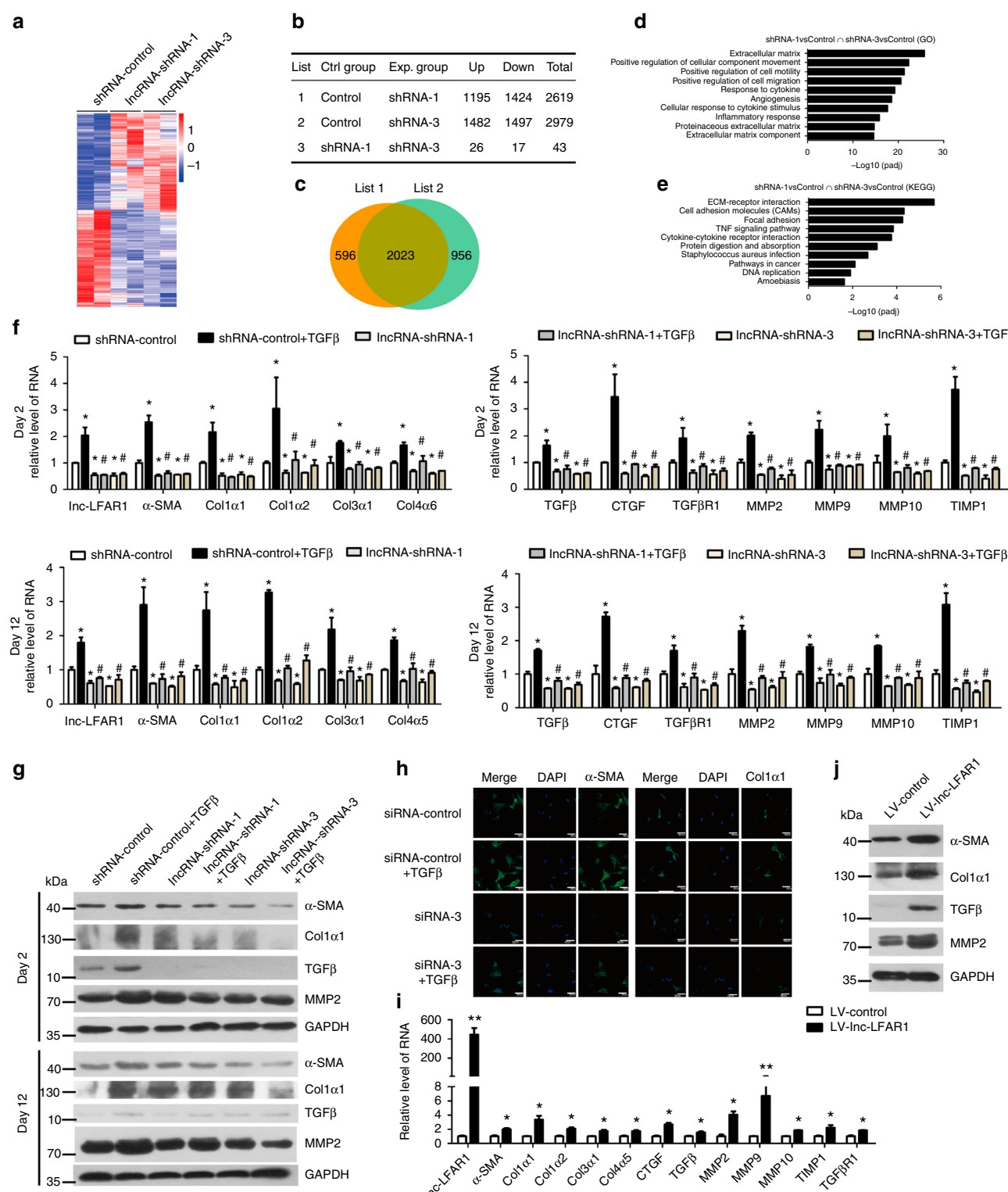

infected with lnc-LFAR1-shRNAs express a lower level of *α-SMA, Col1α1, Col1α2, Col3α1, Col4α5, TGFβ, CTGF, TGFβR1, MMP2/ 9/10* and *TIMP1*, compared with the cells infected with shRNA-control virus (Fig. 3f). Moreover, knockdown of lnc-LFAR1 dramatically decreased TGFβ-induced upregulation of these fibrosis-related genes in primary HSCs. These findings were further confirmed by western blot (Fig. 3g). In addition, confocal microscopy demonstrated that the expressions of both α-SMA and Col1α1 were decreased when HSCs were transfected with lnc-LFAR1 siRNA, simultaneously the TGFβ-induced upregulation of α-SMA and Col1α1 were also greatly blocked (Fig. 3h). To further investigate the roles of lnc-LFAR1 in regulating HSC activation, we tested over-expression of lnc-LFAR1 on primary HSCs (Fig. 3i). Forced expression of lnc-LFAR1 obviously increases the expression of α-SMA, Col1α1, TGFβ and MMP2 assessed by qRT-PCR (Fig. 3i) and western blot (Fig. 3j). Taken together, these results suggest that lnc-LFAR1 promotes the expression of pro-fibrogenic genes and the activation of HSCs.

**Knockdown of Lnc-LFAR1 reduces TGFβ-induced HCs apoptosis.** HC apoptosis triggers HSCs activation either directly by the phagocytosis of the apoptotic bodies, or indirectly by the generation of damage-associated molecular patterns[19]. To investigate the role of lnc-LFAR1 in HCs, we firstly knocked down lnc-LFAR1 and confirmed that lnc-LFAR1 RNAi was effective (Supplementary Fig. 5a), and subsequently, AML12 cells or primary HCs that had been infected with lnc-LFAR1-shRNAs or shRNA-control virus were treated with TGFβ. The results showed that the expressions of pro-fibrogenic genes and pro-inflammation genes were upregulated upon TGFβ treatment. However, knockdown of lnc-LFAR1 abrogates TGFβ-induced upregulation of these genes in AML12 cells (Supplementary Fig. 5b–e) and primary HCs (Supplementary Fig. 6a–d). These findings were further confirmed by confocal microscopy, as demonstrated that the expressions of both Col1α1 and TGFβ were decreased when AML12 cells were transfected with lnc-LFAR1 siRNA, simultaneously the TGFβ-induced upregulation of Col1α1 and TGFβ were also greatly blocked (Supplementary Fig. 6e). On the other hand, over-expression of lnc-LFAR1 increases pro-fibrogenic genes expression in AML12 cells assessed by qRT-PCR and western blot (Supplementary Fig. 7a–d).

We then investigated the effects of lnc-LFAR1 on HC apoptosis. We found TGFβ stimulation dramatically increases AML12 apoptosis. However, lnc-LFAR1 silencing significantly suppresses TGFβ-induced AML12 apoptosis, as indicated by the results of FACS analysis (Supplementary Fig. 8a). Furthermore,

our data revealed that knockdown of lnc-LFAR1 blunt TGFβ-induced dysregulation of apoptotic-related genes in AML12 cells and primary HCs (Supplementary Fig. 8b–d). Taken together, our results suggest that knockdown of lnc-LFAR1 reduces TGFβ-induced HCs apoptosis.

**Lnc-LFAR1 silencing inhibits CCl4- and BDL-induced fibrosis.** To explore the role of lnc-LFAR1 in liver fibrosis in vivo, lenti-shLFAR1 or lenti-NC was intravenously injected into CCl4-treated mice via the tail vein 2 weeks after the first injection of CCl4[20]. Lnc-LFAR1 silencing was confirmed by qRT-PCR in both whole liver extracts and primary HSCs (Supplementary Fig. 9a, b). After 6 weeks of CCl4 treatment, we performed mRNA microarrays to analyze the effect of lnc-LFAR1 downregulation on CCl4-induced liver fibrosis (Fig. 4a). Our data revealed that 1598 mRNAs were upregulated and 620 mRNAs were downregulated in the CCl4-treated mice infected with lenti-NC, while only 292 mRNAs were up-regulated and 112 mRNAs were downregulated in the CCl4-treated mice infected with lenti-shLFAR1. Moreover, there are 133 mRNAs were upregulated and 178 mRNAs were downregulated in the CCl4-treated mice infected with lenti-shLFAR1, compared with the CCl4-treated mice infected with lenti-NC (Fig. 4b). Among these, 234 out of 311 genes were in common with the set of genes deregulated in the CCl4-treated mice infected with lenti-NC (Fig. 4c). In addition, the GO and KEGG pathway analysis revealed that lnc-LFAR1 silencing affects a list of genes associated with collagen fibril organization and TGFβ receptor signaling pathway (Fig. 4d, e and Supplementary Fig. 9c, d). To further examine whether in vivo lnc-LFAR1 down-regulation ameliorates liver fibrosis, we determined the extent of liver fibrosis in lentivirus-infected mice. The CCl4-treated mice infected with lenti-NC developed severe liver fibrosis. However, administration of lenti-shLFAR1 greatly reduced CCl4-induced liver fibrosis, as demonstrated by macroscopic examination, H&E staining, Sirius red staining, TUNEL staining, IHC and western blot for α-SMA and collagen1 (Fig. 4f, g and Supplementary Fig. 9e). The hepatic hydroxyproline content (Fig. 4h) and the serum level of ALT and AST in lenti-shLFAR1-infected mice were also significantly decreased in comparison with the CCl4-treated mice infected with lenti-NC (Supplementary Table 1). In addition, lentivirus-mediated knockdown of lnc-LFAR1 resulted in significant reduced expression of pro-fibrogenic (*α-SMA, Col1α1, Col1α2, CTGF, MMP2/9* and *TIMP1*) (Fig. 4i), pro-inflammation (*TNFα, IL1β* and *MCP1*) and pro-apoptosis (*Bax* and *BAD*) genes (Fig. 4j). Similarly, we demonstrated that the expression of these pro-fibrogenic, pro-inflammation and pro-apoptosis genes in the HSCs (Supplementary Fig. 9f, g) and HCs (Supplementary

**Fig. 3** Lnc-LFAR1 regulates the expression of extracellular matrix genes in primary HSCs. **a** Microarray heat map demonstrates clustering of primary HSCs infected with shRNA-control (*n* = 2), lncRNA-shRNA1 (*n* = 2) and lncRNA-shRNA3 (*n* = 2). Hierarchical cluster analysis of significantly differentially expressed mRNAs: *bright blue*, under-expression; *gray*, no change; *bright red*, over-expression. **b–e** Microarray analyses, such as differential screening **b**, **c**, GO **d**, and KEGG pathway analysis **e**, were performed. **f**, **g** Primary HSCs at day 2 or day 12 were infected with two separated lentivirus-mediated shLFAR1 for 72 h and further treated with 10 ng ml⁻¹ TGFβ for additional 24 h. The expression of *lnc-LFAR1, α-SMA, Col1α1, Col1α2, Col3α1, Col4α5, TGFβ, CTGF, TGFβR1, MMP2/9/10* and *TIMP1* was detected by qRT-PCR **f**. The protein levels of α-SMA, Col1α1, TGFβ and MMP2 were detected by western blot. GAPDH was used as an internal control **g**. **h** Primary HSCs were transfected with siRNA for lnc-LFAR1 for 48 h and further treated with 10 ng ml⁻¹ TGFβ for additional 24 h. The expression of α-SMA and Col1α1 was determined by confocal microscopy. DAPI-stained nuclei blue; scale bar, 50 μm. **i** The RNA levels of *lnc-LFAR1, α-SMA, Col1α1, Col1α2, Col3α1, Col4α5, TGFβ, CTGF, TGFβR1, MMP2/9/10* and *TIMP1* were detected in primary HSCs infected with lenti-lnc-LFAR1 or lenti-control by qRT-PCR. **j** The protein levels of α-SMA, Col1α1, TGFβ and MMP2 were detected in lnc-LFAR1 over-expressed primary HSCs by western blot. GAPDH was used as an internal control. Uncropped blots of this figure accompanied by the location of molecular weight markers are shown in Supplementary Fig. 18. In **f** and **i**, the number of biological replicates for each experiment was *n* ⩾ 3. Data are presented as means ± s.e.m. *P* values were analyzed by one-way analysis of variance followed by *post hoc* comparison in **f**, and by Student's *t*-test in **i**. */#*P* < 0.05, **/##*P* < 0.01. **P* < 0.05 vs shRNA-control, #*P* < 0.05 vs shRNA-control + TGFβ in **f**; and **P* < 0.05 vs LV-control in **i**

Fig. 9h, i) isolated from lenti-shLFAR1-infected mice exhibits a marked decrease in comparison with the lenti-NC mice. To exclude the possibility that lnc-LFAR1 alters the metabolism or toxicity of $CCl_4$ rather than by altering stellate cell responses, we confirmed the results in a BDL-induced mice liver fibrosis model. Two separated lenti-shLFAR1 or lenti-NC was intravenously injected into BDL-treated mice via the tail vein 1 day before the surgical operation. The extent of liver fibrosis in lentivirus-infected mice was determined by H&E staining, Sirius red staining, TUNEL staining, IHC and western blot for α-SMA and collagen1 (Supplementary Fig. 10a, b), the hepatic hydroxyproline content (Supplementary Fig. 10c) and the serum level of ALT and

AST (Supplementary Table 2). Moreover, qRT-PCR of pro-fibrogenic, pro-inflammation and pro-apoptosis genes in liver tissues (Supplementary Figs 10d, e and 11a), primary HSCs (Supplementary Fig. 11b) and primary HCs (Supplementary Fig. 11c, d) from lenti-shLFAR1-infected mice exhibits a marked decrease in comparison with the lenti-NC mice. Taken together, our results suggest that knockdown of lnc-LFAR1 attenuates $CCl_4$- and BDL-induced liver fibrosis in vivo.

**Smad2/3 mediates TGFβ-induced Lnc-LFAR1 expression.** Based on the dysregulation of lnc-LFAR1 during liver fibrogenesis, we further characterized the mechanisms involved in this

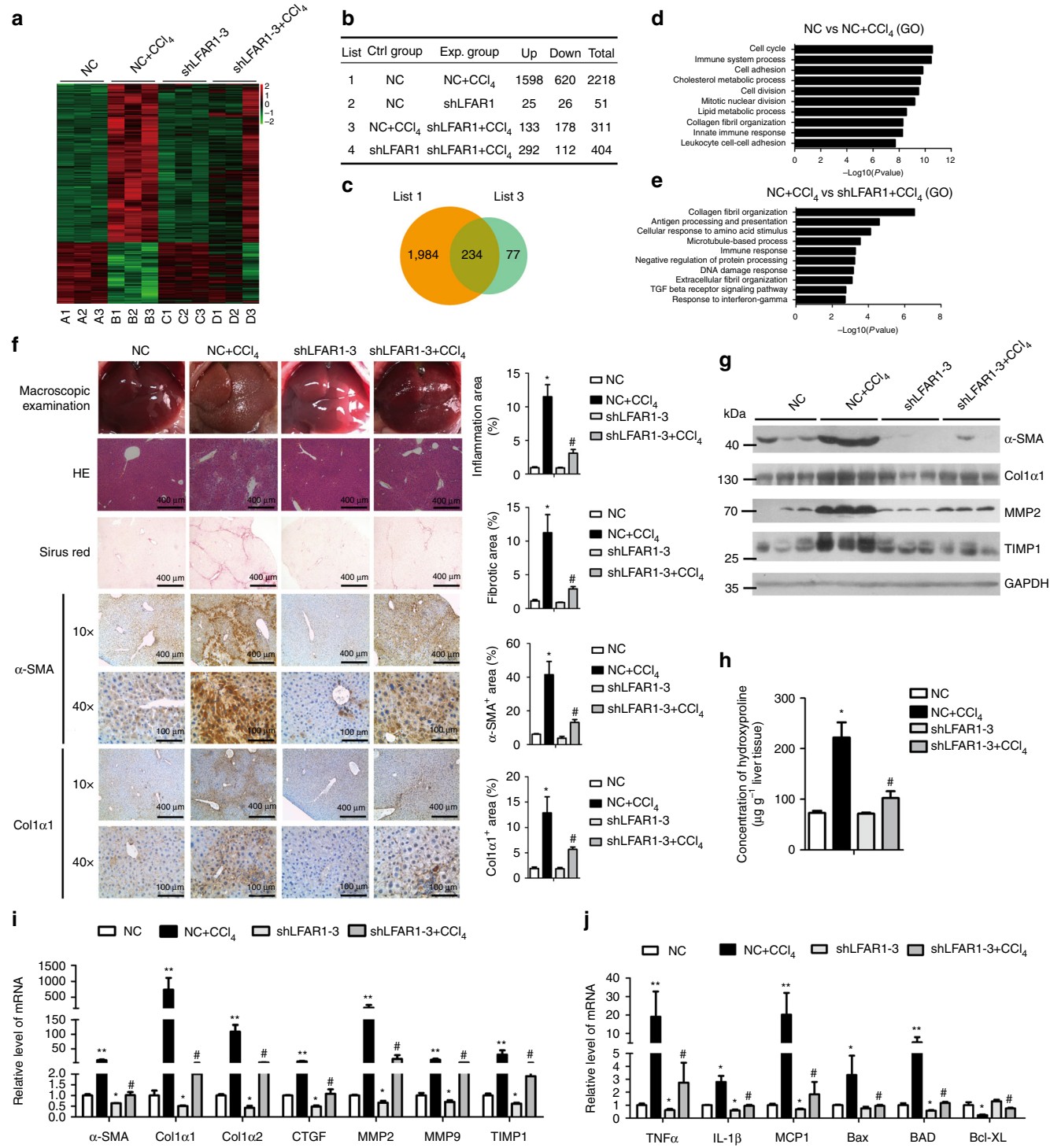

dysregulation. We have demonstrated that TGFβ promotes lnc-LFAR1 expression in HSCs and HCs (Fig. 2e, f and Supplementary Fig. 4b, c). Since TGFβ and the downstream mediators, Smad2/3, represent key pro-fibrogenic mediators and we therefore tested whether TGFβ might regulate the expression of lnc-LFAR1 through Smad2/3. Firstly, Smad2/3 expression was silenced by two separated shRNA-Smad2 or shRNA-Smad3 virus, respectively, and the silencing effects were confirmed by qRT-PCR and western blot in HSCs (Fig. 5a, b) and AML12 cells (Supplementary Fig. 12a). The expression of lnc-LFAR1 was significantly reduced in cells infected with shRNA-Smad2 or shRNA-Smad3 virus (Fig. 5a and Supplementary Fig. 12a). Secondly, chromatin immunoprecipitation (ChIP) further showed that the promoter region of lnc-LFAR1 and PAI, a Smad2/3 target gene as positive control, but not GAPDH, are immunoprecipitated by the Smad2/3 antibody, suggesting that endogenous Smad2/3 directly binds to the lnc-LFAR1 promoter. Moreover, the promoter occupancies were increased upon TGFβ treatment (Fig. 5c, d and Supplementary Fig. 12b, c). Finally, three potential Smad2/3 binding sites (SBE) were predicted in the lnc-LFAR1 promoter region by using ALGGEN-PROMO and JASPAR (Fig. 5e). Lnc-LFAR1 promoter fragments containing SBE or mutant SBE (Mut1:-882/-871 and Mut2:-1291/-1274) were cloned into the vector PGL3-basic to transfect HSCs or AML12 cells along with the treatment with 10 ng ml$^{-1}$ TGFβ and the promoter activity was measured by luciferase reporter gene assay 72 h post transfection. The results showed the lnc-LFAR1 promoter activity was apparently enhanced by TGFβ. However, mutation of the SBE1 (Mut1:-882/-871) partially abrogates the TGFβ response, whereas mutation of the SBE2 (Mut2:-1291/-1274) completely abrogates the TGFβ response (Fig. 5e, f and Supplementary Fig. 12d, e), suggesting that the major binding site locates in -1291/-1274. Taken together, these results indicate that TGFβ promotes Smad2/3 to bind the promoter of lnc-LFAR1, and subsequently increases its expression.

**Lnc-LFAR1 promotes Smad2/3 expression and phosphorylation.** Since the TGFβ pathway is one of the well-investigated signaling cascades in liver fibrosis and the GO and KEGG pathway analysis revealed that lnc-LFAR1 silencing affects a list of genes associated with TGFβ receptor signaling pathway (Fig. 4d, e and Supplementary Fig. 9c, d), we were interested in the functional role of lnc-LFAR1 involved in the TGFβ pathway. We measured the level of total Smad2/3 and phosphorylated Smad2/3 in lnc-LFAR1 downregulated HSCs and lnc-LFAR1 over-expressed HSCs. Lnc-LFAR1 silencing not only decreases

the basal level of phosphorylated Smad2/3 but also inhibits TGFβ-induced Smad2/3 phosphorylation (Fig. 6a). Interestingly, we also found that both the protein and mRNA levels of total Smad2/3 are decreased in lnc-LFAR1 downregulated HSCs, whereas increased in lnc-LFAR1 over-expressed HSCs (Fig. 6a–c). Similar results were observed in AML12 cells (Supplementary Fig. 13a–c). Moreover, the results confocal microscopy with HSCs (Fig. 6d) and AML12 cells (Supplementary Fig. 13d) showed that TGFβ increases the level of phosphorylated Smad2/3 and promotes the translocation of Smad2/3 from the cytoplasm to the nucleus, while knockdown of lnc-LFAR1 dramatically decreases TGFβ-induced Smad2/3 phosphorylation and translocation. In order to determine whether lnc-LFAR1 promotes Smad2/3 phosphorylation independent of the increased level of total Smad2/3, SB431542, a potent and selective inhibitor of TGFβR1 which is the major receptor for canonical signaling of TGFβ, was used to treat lnc-LFAR1 over-expressed primary HSCs and AML12 cells. The results showed that SB431542 eliminates the phosphorylation induced by lnc-LFAR1 (Fig. 6e and Supplementary Fig. 13e). These findings were also confirmed by two separated TGFβR1 siRNAs (Fig. 6f and Supplementary Fig. 13f). Moreover, RIP revealed that lnc-LFAR1, but not lnc-ENSMUST00000154817 and Actin, directly associates with TGFβR1 (Fig. 6g). Co-immunoprecipitation confirmed that over-expression of lnc-LFAR1 promotes the association of TGFβR1 with Smad2/3 in primary HSCs and AML12 cells (Fig. 6h and Supplementary Fig. 13g), suggesting that lnc-LFAR1 promotes Smad2/3 phosphorylation through TGFβR1. In addition, we validated the differential expression of the key TGFβ pathway genes in livers from lenti-shLFAR1- or lenti-NC-infected mice treated with or without CCl$_4$ or BDL. As shown in Supplementary Fig. 14a and 15a, lentivirus-mediated knockdown of lnc-LFAR1 results in reduced mRNA level of *TGFβ, TGFβR1, Smad2* and *Smad3,* and inhibits CCl$_4$- and BDL-induced upregulation of these genes. Furthermore, knockdown of lnc-LFAR1 leads to a decreased protein level of TGFβ, phosphorylated Smad2/3 and total Smad2/3, and dramatically abrogates CCl$_4$- and BDL-induced upregulated protein levels shown by western blot (Supplementary Figs 14b and 15b). These findings were also confirmed in HSCs and HCs isolated from the lentivirus-injected mice (Supplementary Figs 14c, d and 15c, d). In addition, IHC demonstrated that lnc LFAR1 silencing significantly decreases TGFβ expression, and the CCl$_4$- and BDL-upregulated TGFβ was simultaneously greatly blocked in the lenti-shLFAR1-injected mice (Supplementary Figs 14e and 15e). Taken together, these data suggest that lnc-LFAR1 up-regulates the expression of Smad2/3 and promotes its phosphorylation in liver fibrogenesis.

---

**Fig. 4** Knockdown of lnc-LFAR1 attenuates CCl$_4$-induced liver fibrosis in vivo. Mice were treated with oil in combination with injection of lenti-NC (negative control, n = 10), or CCl$_4$ in combination with injection of lenti-NC (NC+CCl$_4$, n = 10), or oil in combination with injection of lenti-shLFAR1-3 (shLFAR1-3, n = 10), or CCl$_4$ in combination with injection of lenti-shLFAR1-3 (shLFAR1-3+CCl$_4$, n = 10). **a** Microarray heat map demonstrates clustering of NC (n = 3), NC+CCl$_4$ (n = 3), shLFAR1-3 (n = 3) and shLFAR1-3+CCl$_4$ (n = 3) Balb/c-mice. Hierarchical cluster analysis of significantly differentially expressed mRNAs: *bright green*, under-expression; *gray*, no change; *bright red*, over-expression. **b**–**e** Microarray analyses, such as differential screening **b**, **c** and GO analysis **d**, **e**, were performed between the four groups. **f** Liver fibrosis was evaluated by macroscopic examination, H&E staining, Sirius red staining and IHC for α-SMA and collagen1. Scale bars, 400 μm for H&E staining, Sirius red staining and IHC (objective, ×10); 100 μm for IHC (objective, ×40). *Right*, five images of each liver and five livers from different mice were quantified for each group. **g** The protein levels of α-SMA, Col1α1, MMP2 and TIMP1 were determined by western blot. GAPDH was used as an internal control. Uncropped blots of this figure accompanied by the location of molecular weight markers are shown in Supplementary Fig. 18. **h** Quantification of hepatic hydroxyproline content. The data are expressed as hydroxyproline (μg) per liver wet weight (g) (n = 7 per group). **i,j** The mRNA levels of hepatic pro-fibrogenic genes (α-SMA, Col1α1, Col1α2, CTGF, MMP2/9 and TIMP1) (**i**), pro-inflammation genes (TNFα, IL1β and MCP1) and apoptosis-related genes (Bax and BAD) (**j**) were determined by qRT-PCR. In **i** and **j**, the number of biological replicates for each experiment was n ⩾ 5. Data are presented as means ± s.e.m. P values were analyzed by one-way analysis of variance followed by *post hoc* comparison in **f**, **h**–**j**. */#P < 0.05, **/##P < 0.01. *P < 0.05 vs NC, #P < 0.05 vs NC+CCl$_4$

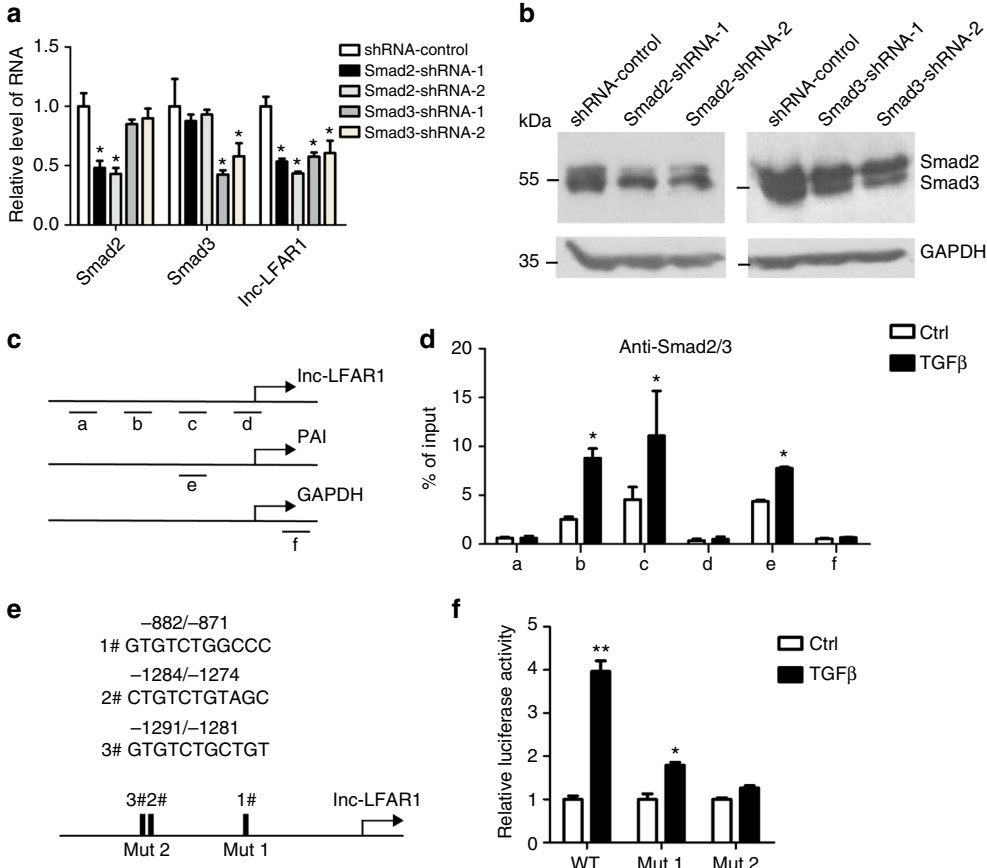

**Fig. 5** Smad2/3 mediates TGFβ-induced lnc-LFAR1 expression in HSCs. **a**, **b** Relative expression levels of Smad2, Smad3 and lnc-LFAR1 in primary HSCs infected with lenti-shSMAD2 or lenti-shSMAD3 or lenti-control virus were examined by qRT-PCR **a** and western blot **b** compared to GAPDH. Uncropped blots of this figure accompanied by the location of molecular weight markers are shown in Supplementary Fig. 18. **c**, **d** ChIP analyses of primary HSCs treated with or without 10 ng ml$^{-1}$ TGFβ for 24 h were conducted on lnc-LFAR1 (primer set **a**–**d**), PAI (the positive control; primer **e**) and GAPDH (the negative control; primer **f**) promoter regions using anti-Smad2/3 antibody. Enrichment was shown relative to input. **e** Diagram of the predicted three Smad2/3 binding sites in the lnc-LFAR1 promoter region. Points mutation of binding site 1 (Mut1), and binding sites 2 and 3 (Mut2) of Smad2/3 were indicated. **f** Luciferase analysis. Primary HSCs were transfected with the luciferase reporter constructs harboring either Smad2/3 binding sites or the mutated binding sites for 48 h, and further treated with 10 ng ml$^{-1}$ TGFβ for additional 24 h. The cells were lysed for dual luciferase analysis. The Renilla was transfected as an internal control. In **a**, **d** and **f**, the number of biological replicates for each experiment was $n \geqslant 3$. Data are presented as means ± s.e. m. $P$ values were analyzed by Student's $t$-test. *$P < 0.05$. *$P < 0.05$ vs shRNA-control in **a**; and *$P < 0.05$ vs Ctrl in **d** and **f**

**Lnc-LFAR1 activates notch pathway to promote liver fibrosis.** To investigate whether lnc-LFAR1 also affects other liver fibrosis-related pathways with the exception of TGFβ pathway, we detected these pathways components or target genes, including Wnt target genes Cyclin D1 and Myc[21], Hippo target genes Ankrd1 and Areg[22], Notch signaling-related molecules and target genes Notch2, Notch3, Hes1 and Hey2 and Hedgehog target genes Ptch1 and Gli1[23,24], in lnc-LFAR1 downregulated HSCs. As shown in Figs 7a, b, both the protein and mRNA levels of Notch2, Notch3, Hes1 and Hey2 were decreased in lnc-LFAR1 down-regulated HSCs, whereas increased in lnc-LFAR1 over-expressed HSCs (Figs 7c, d). These results were further confirmed in AML12 cells (Supplementary Fig. 16a–d). In addition, we also found that the expression of Notch2, Notch 3, Hes1 and Hey2 are significantly increased in the CCl$_4$- and BDL-treated mice, compared with the mice only infected with lenti-NC (Fig. 7e and Supplementary Fig. 16e). Moreover, lentivirus-mediated knock-down of lnc-LFAR1 resulted in reduced expression of Notch2, Notch 3, Hes1 and Hey2, and inhibits CCl$_4$- and BDL-induced upregulation of these genes (Fig. 7e and Supplementary Fig. 16e). We also confirmed these findings in HSCs and HCs isolated from the lentivirus-injected mice and obtained similar results

(Supplementary Fig. 16f–i). Consistently, western blot (Fig. 7f and Supplementary Fig. 16j) and IHC (Fig. 7g and Supplementary Fig. 16k) revealed that knockdown of lnc-LFAR1 reduces Notch2, Notch 3 and Hes1 expression and inhibits CCl$_4$- and BDL-increased expression of these genes. Taken together, these data show that lnc-LFAR1 promotes liver fibrogenesis and HSCs activation by activating Notch pathway.

**Lnc-LFAR1 binds Smad2/3 to regulate target genes expression.** We next sought to explore the underlying molecular mechanism by which lnc-LFAR1 promotes liver fibrogenesis. Recently, many lncRNAs have been reported to function as competing endogenous RNAs (ceRNA) by competitively binding common microRNAs in the cytoplasm or physically associate with specific proteins such as chromatin remodeling complex and transcription factors in the nucleus[7]. Because the distribution of lnc-LFAR1 in HSCs was located in both cytoplasm and nucleus, we firstly investigated whether lnc-LFAR1 associates with the AGO2 protein, a key component of the microRNA-containing RISC complex. However, RIP revealed that lnc-LFAR1 was not associated with the AGO2 protein (Supplementary Fig. 17a). As it has

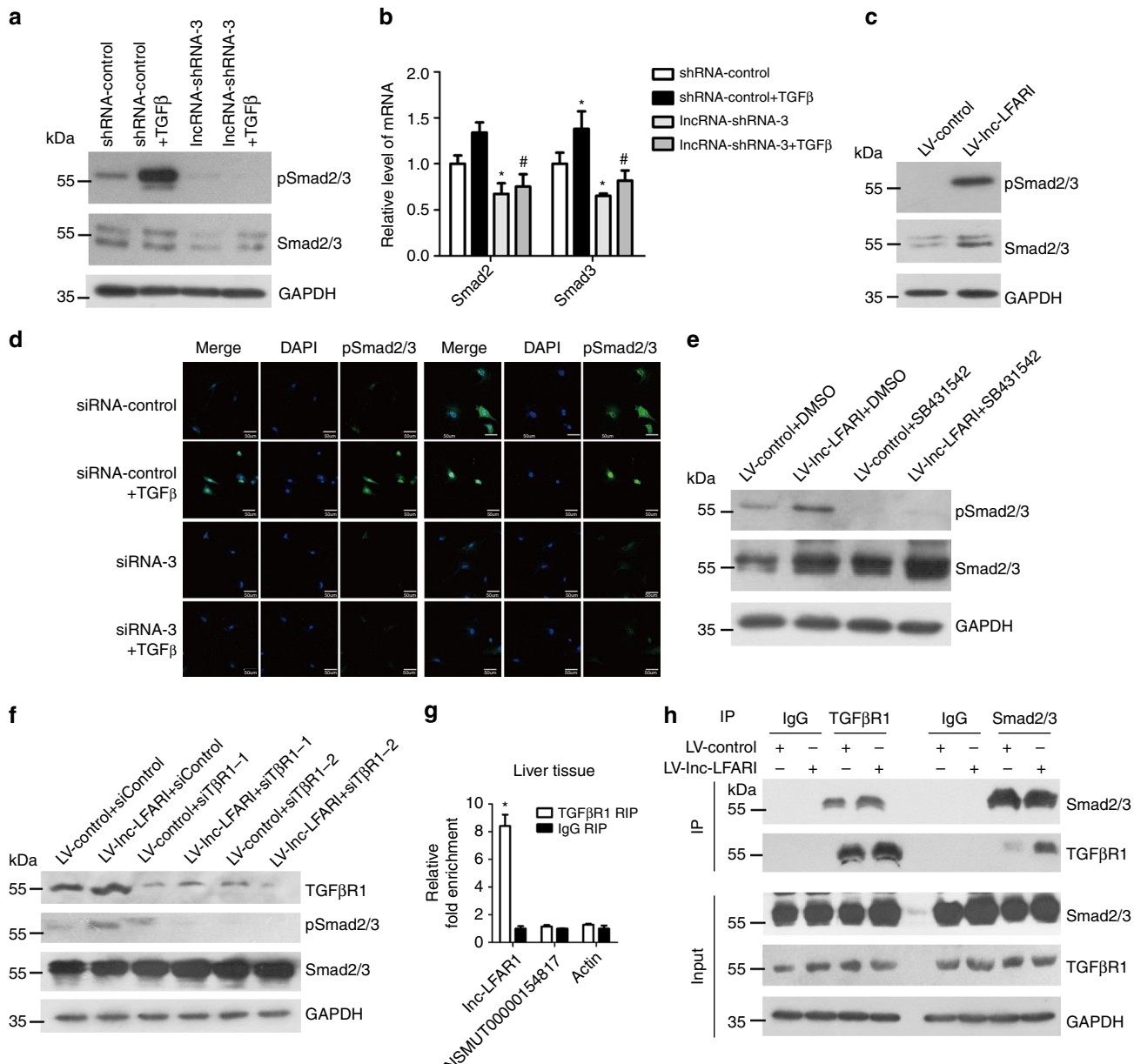

**Fig. 6** Lnc-LFAR1 upregulates the expression of Smad2/3 and promotes its phosphorylation in liver fibrogenesis. **a**, **b** Primary HSCs were infected with lentivirus-mediated shLFAR1 for 72 h and further treated with 10 ng ml$^{-1}$ TGFβ for additional 24 h. pSmad2/3 and total Smad2/3 levels were detected by western blot **a**. *Smad2* and *Smad3* mRNA levels were determined by qRT-PCR **b**. **c** pSmad2/3 and total Smad2/3 levels were detected in lnc-LFAR1 over-expressed primary HSCs by western blot. GAPDH was used as an internal control. **d** Primary HSCs were transfected with siRNA for lnc-LFAR1 for 48 h and further treated with 10 ng ml$^{-1}$ TGFβ for additional 24 h. The expression and location of pSmad2/3 and total Smad2/3 were determined by confocal microscopy. DAPI-stained nuclei blue; scale bar, 50 μm. **e** Primary HSCs were infected with LV-lnc-LFAR1 for 72 h and further treated with TGFβR1 inhibitor SB431542 for additional 48 h. pSmad2/3 and total Smad2/3 levels were detected by western blot. GAPDH was used as an internal control. **f** Primary HSCs were infected with LV-lnc-LFAR1 for 72 h and further transfected with siRNAs for TGFβR1 for additional 48 h. TGFβR1, pSmad2/3 and total Smad2/3 levels were detected by western blot. GAPDH was used as an internal control. **g** qRT-PCR detection of lnc-LFAR1, lncRNA-ENSMUST00000154817 and Actin retrieved by TGFβR1-specific antibody compared with IgG in the RIP assay within the single-cell suspensions isolated from mouse liver. **h** TGFβR1 and Smad2/3 antibodies were used for co-immunoprecipitation (IP) with primary HSCs lysates infected with or without LV-lnc-LFAR1. Uncropped blots of this figure accompanied by the location of molecular weight markers are shown in Supplementary Fig. 18. In **b** and **g**, the number of biological replicates for each experiment was *n* ⩾ 3. Data are presented as means ± s.e.m. *P* values were analyzed by one-way analysis of variance followed by *post hoc* comparison in **b**, and by Student's *t*-test in **g**. */#*P* < 0.05. *P* < 0.05 vs shRNA-control, #*P* < 0.05 vs shRNA-control + TGFβ in **b**; and *P* < 0.05 vs IgG RIP in **g**

been reported that 20% of all lncRNAs physically associate with the Polycomb Repressive Complex 2 (PRC2)[7], we next performed RIP assay to pull down endogenous RNAs associated with SUZ12, an important subunit of PRC2, Unfortunately, we only observed a significant enrichment of lnc-MALAT1 as positive control but no enrichment of β-actin or lnc-LFAR1 with the SUZ12 antibody, compared with IgG (Supplementary Fig. 17b). In addition, we immunoprecipitated the transcription factor Smad2/3 with

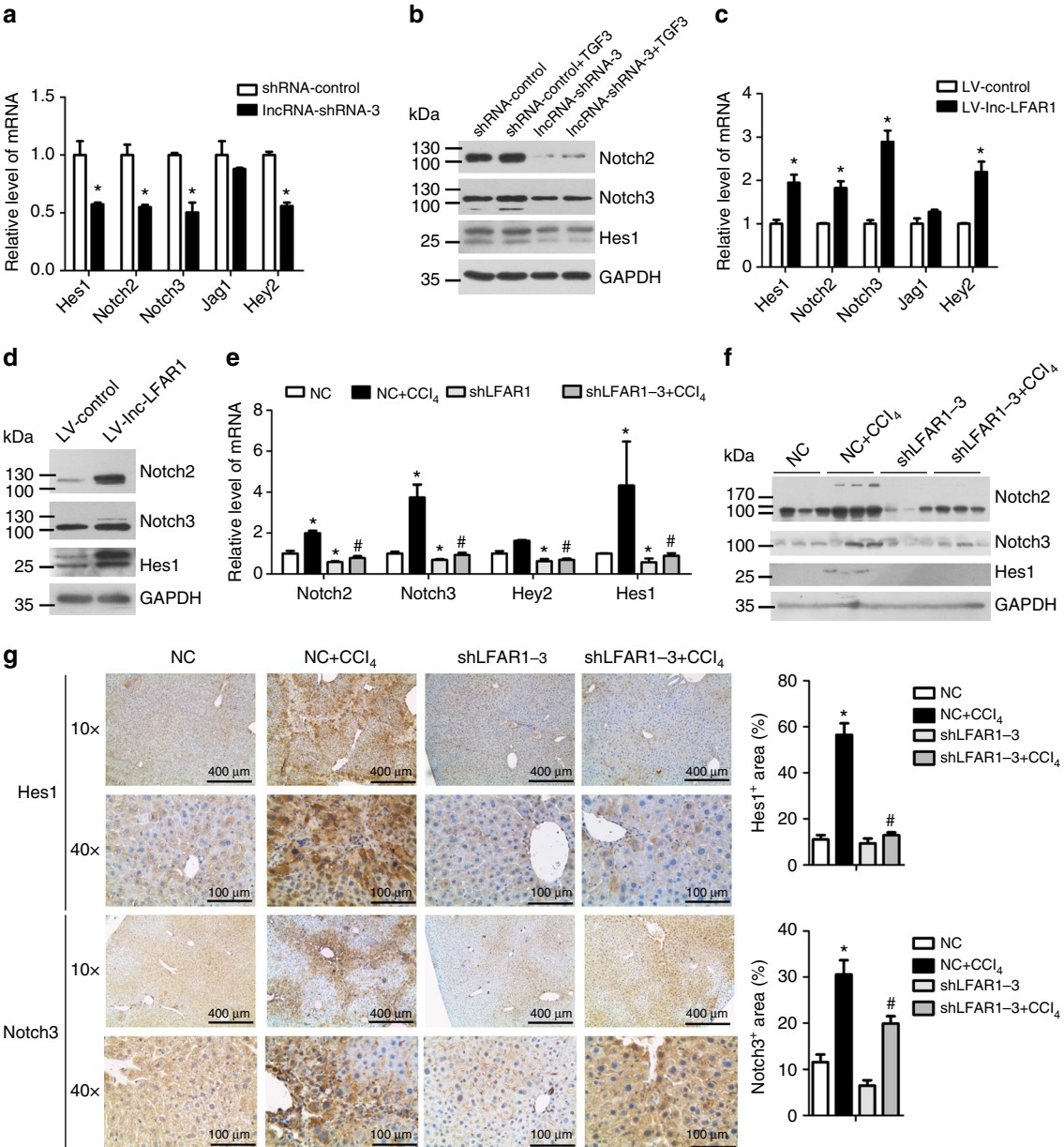

**Fig. 7** Lnc-LFAR1 promotes liver fibrosis through activating Notch signaling in primary HSCs. **a** The mRNA levels of *Hes1, Notch2, Notch3, Jag1* and *Hey2* were detected in lnc-LFAR1 downregulated primary HSCs by qRT-PCR. **b** Primary HSCs were infected with lentivirus-mediated shLFAR1 for 72 h and further treated with 10 ng ml$^{-1}$ TGFβ for additional 24 h. The protein levels of Nocth2, Notch3 and Hes1 were detected by western blot. GAPDH was used as an internal control. **c**, **d** The expression of *Hes1, Notch2, Notch3, Jag1* and *Hey2* was detected in lnc-LFAR1 over-expressed primary HSCs by qRT-PCR (**c**) and western blot **d**. GAPDH was used as an internal control. **e**, **f** Mice were treated with oil in combination with injection of lenti-NC (NC, *n* = 10), or CCl$_4$ in combination with injection of lenti-NC (NC+CCl$_4$, *n* = 10), or oil in combination with injection of lenti-shLFAR1 (shLFAR1, *n* = 10), or CCl$_4$ in combination with injection of lenti-shLFAR1 (shLFAR1+CCl$_4$, *n* = 10). The mRNA and protein levels of Hes1, Notch2, Notch3 and Hey2 were determined in liver tissues by qRT-PCR **e** and western blot **f**. GAPDH was used as an internal control. **g** Notch3 and Hes1 levels were detected by IHC. Scale bars, 400 μm for IHC (objective, ×10); 100 μm for IHC (objective, ×40). *Right*, five images of each liver and five livers from different mice were quantified for each group. Uncropped blots of this figure accompanied by the location of molecular weight markers are shown in Supplementary Fig. 18. In **a**, **c** and **e**, the number of biological replicates for each experiment was *n* ≥ 3. Data are presented as means ± s.e.m. *P* values were analyzed by Student's *t*-test in **a** and **c**, and by one-way analysis of variance followed by *post hoc* comparison in **e** and **g**. */#*P* < 0.05. *P* < 0.05 vs shRNA-control in **a**; and *P* < 0.05 vs LV-Control in **c**; and *P* < 0.05 vs NC, #*P* < 0.05 vs NC+CCl$_4$ in **e** and **g**

specific antibody from lysates of both the liver tissue (Fig. 8a) and lnc-LFAR1 over-expressed AML12 cells (Supplementary Fig. 17c). The results showed a significant enrichment of lnc-LFAR1 but no enrichment of β-actin with the Smad2/3 antibody, compared with the IgG antibody, suggesting that lnc-LFAR1 physically interacts with Smad2/3. To further examine whether

lnc-LFAR1 increases the expression of α-SMA and Col1α1 via binding with Smad2/3, we knocked down Smad2 and Smad3 in lnc-LFAR1 over-expressed HSCs (Fig. 8b). The knockdown of Smad2 or Smad3 in lnc-LFAR1 over-expressing cells abrogates lnc-LFAR1-increased α-SMA and Col1α1 protein level. In addition, to address whether lnc-LFAR1 regulates the transcription of

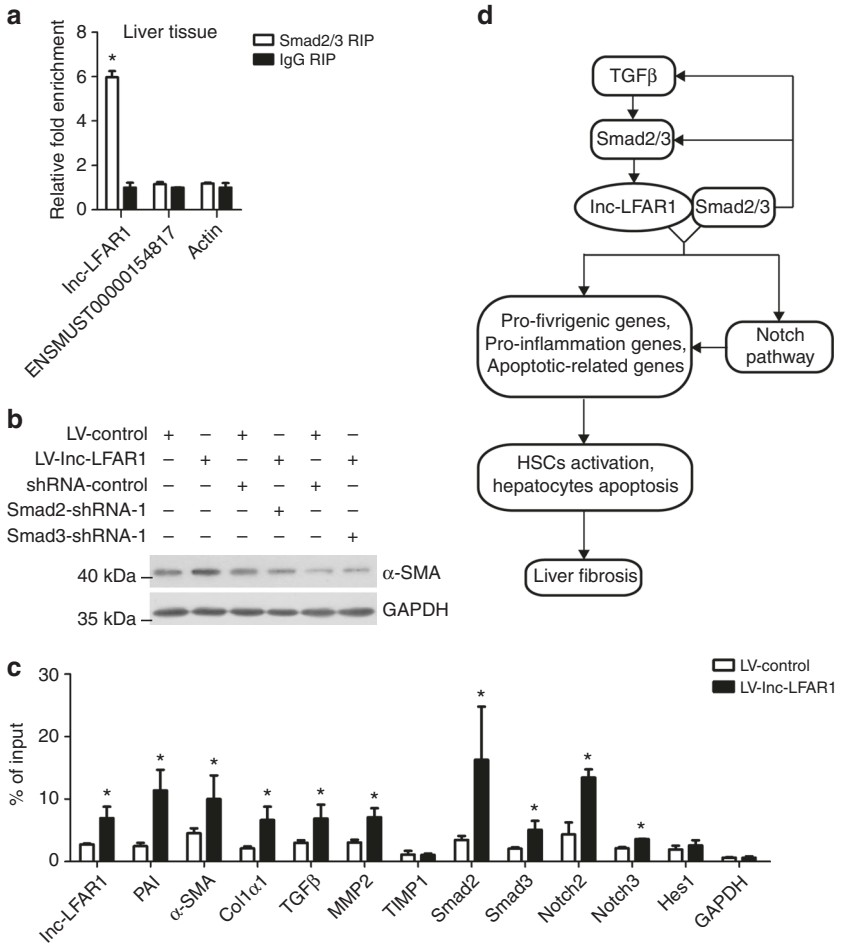

**Fig. 8** Lnc-LFAR1 interacts with Smad2/3 to regulate target genes expression. **a** qRT-PCR detection of lnc-LFAR1, lncRNA-ENSMUST00000154817 and actin retrieved by Smad2/3-specific antibody compared with IgG in the RIP assay within the single-cell suspensions isolated from mouse liver. **b** The protein level of α-SMA in lnc-LFAR1 over-expressed primary HSCs simultaneously infected with leni-shSmad2 or lenti-shSmad3 virus was determined by western blot. GAPDH was used as an internal control. Uncropped blots of this figure accompanied by the location of molecular weight markers are shown in Supplementary Fig. 18. **c** Primary HSCs were infected with lenti-lnc-LFAR1 or lenti-control, and ChIP analyses were performed on indicated genes promoter regions using anti-Smad2/3 antibody. Enrichment was shown relative to input. **d** Schematic representation of the TGFβ/Smad2/3/lnc-LFAR1 pathway and its function in the progression of liver fibrosis. In **a** and **c**, the number of biological replicates for each experiment was $n \geqslant 3$. Data are presented as means ± s.e.m. P values were analyzed by Student's t-test. *$P < 0.05$. *$P < 0.05$ vs IgG RIP in **a**; and *$P < 0.05$ vs LV-control in **c**

its target genes through promoting the binding of Smad2/3 to the promoters, ChIP analysis was performed in primary HSCs and AML12 cells infected with lenti-LFAR1 or lenti-control virus. The data demonstrated that lnc-LFAR1 increases the promoter occupancy of Smad2/3 to the target genes, including PAI, α-SMA, Col1α1, TGFβ, MMP2, Smad2, Smad3, Notch2, Notch3 and lnc-LFAR1 itself. However, there were no increased binding in the TIMP1 and Hes1 promoters (Fig. 8c and Supplementary Fig. 17d), indicating that lnc-LFAR1 may regulate the expression of these genes in other ways. Taken together, our data demonstrate that lnc-LFAR1 interacts with Smad2/3 and promotes the binding of these factors to the promoters of the target genes.

## Discussion

Various cytokines, growth factors and miRNAs have been shown to regulate the genes that orchestrate activation, apoptosis and proliferation in liver fibrogenesis[1, 2]; however, little information exists on the lncRNAs that regulate liver fibrogenesis. In this study, we demonstrate that several lncRNAs are specifically regulated in mouse models of liver fibrosis through lncRNA microarray analysis, leading to the identification of a

liver-enriched lnc-LFAR1, which could be regulated by TGFβ. Our data showed that lnc-LFAR1 functions as a positive regulator in liver fibrosis both in vitro and in vivo. Mechanistically, our results indicated that lnc-LFAR1 promotes the association of Smad2/3 with TGFβR1 which subsequently phosphorylates Smad2/3 in the cytoplasm. Furthermore, lnc-LFAR1 binds directly to Smad2/3, and this association activates both TGFβ and Notch pathways through directly regulating the transcription of TGFβ, Smad2, Smad3, Notch2, Notch3 and Hes1, revealing that TGFβ1/Smad2/3/lnc-LFAR1 pathway might provide a positive feedback loop that augments Smad2/3 response and a novel link connecting TGFβ with Notch pathway (Fig. 8d). All these data support our conclusion that lnc-LFAR1 has pleiotropic effects on HSCs activation, HCs apoptosis and liver fibrogenesis, thus providing support for its use as a potential target of fibrosis.

Although the function of lncRNA is important, the rapid sequence evolution of lncRNAs presents a challenge to identifying functional counterparts between species[5, 6, 25–28]. For example, there is no clear orthologous Bvht, a heart-associated lncRNA, in human or rat genomes, and Bvht represents a lineage-specific non-conserved lncRNA[24]. Moreover, few orthologues of mammalian lncRNAs exist outside of mammals and are highly

variable among species[5, 6, 25–28]. In our study, the full length of lnc-LFAR1 is also poorly conserved across species and a clear human orthologue could not be immediately identified using the BLAST algorithm. However, batch coordinate conversion between mouse and human assemblies revealed lnc-LFAR1 is localized in a region of the mouse genome that is syntonic to human chromosome 4q25 and adjacent to the human CYP2U1 and HADH genes. In the future it will be of interest to determine if a human 'orthologue' of lnc-LFAR1 has evolved in its primary sequence yet maintains its structure or function with the development of the high-throughput sequencing technology and computational analysis.

Liver fibrosis is a result of the wound-healing response of the liver to repeated injury. After an acute liver injury, parenchymal cells regenerate and replace the necrotic or apoptotic cells. If the hepatic injury persists, then eventually the parenchymal cells regeneration fails, leading to increased HC apoptosis, HSCs activation and liver fibrosis[2, 18, 29]. Thus rational treatment approaches for liver fibrosis may include drugs that target HC apoptosis, stellate cell activation, or both. In this study, we found that lnc-LFAR1 is downregulated in the fibrotic livers, while lnc-LFAR1 is significantly increased in primary HSCs extracted from fibrotic mice. As hypothesized, it is likely that the incongruent regulation patterns in other hepatic cell types result in inconsistent overall levels. We then detected the expression of lnc-LFAR1 in primary HCs extracted from fibrotic mice and control mice. Not surprisingly, lnc-LFAR1 expression was downregulated in primary HCs extracted from fibrotic mice. However, our results have shown that lnc-LFAR1 expression is upregulated in primary HCs and AML12 cells upon TGFβ treatment, and knockdown of lnc-LFAR1 reduces TGFβ-induced pro-fibrogenic, pro-inflammation and pro-apoptosis genes expression and apoptosis in HCs (Supplementary Fig. 5–8), suggesting that lnc-LFAR1 promotes apoptosis of HCs. To explain the inconsistent results observed in vivo and in vitro, we investigate the expression of lnc-LFAR1 in liver tissues and primary HCs from mice treated with CCl4 or BDL for various times. Interestingly, we found the level of lnc-LFAR1 is decreased drastically at 2 weeks after CCl4 injection and 3 days after BDL, both are early stage of liver fibrosis, while it is gradually increased with persist injury (Supplementary Fig. 4d–f). At early stage of liver fibrosis parenchymal cells regenerate and replace the necrotic or apoptotic cells. If the hepatic injury persists, then eventually the parenchymal cells regeneration fails, leading to increased HC apoptosis. Although lnc-LFAR1 is decreased in primary HCs extracted from fibrotic mice at 6 weeks, compared with normal control, it is increased compared with that at 2 weeks, indicating the apoptosis of HCs is increased. This could explain the finding that lnc-LFAR1 promotes apoptosis of HCs (Supplementary Fig. 5–8), but its level is still reduced, compared with normal control (Supplementary Fig. 4a), at the stage of 6 weeks after CCl4 injection. In addition, the expression of lnc-LFAR1 was also increased in culture activated primary HSCs and TGFβ-treated HSCs, suggesting lnc-LFAR1 is involved in HSCs activation during the progression of liver fibrosis. These results, overall, showed that lnc-LFAR1 silencing inhibits liver fibrosis through inhibiting the activation of HSCs and repressing the apoptosis of HCs.

Accumulating studies have revealed various pro-fibrogenic pathways involved in liver fibrosis, including TGFβ/Smad, Wnt/β-catenin, Notch, and Hedgehog pathways[18]. Within these, considering the significance of the TGFβ/Smad pathway in regulating fibrogenesis, researchers are trying to block the TGFβ/Smad signal in order to suppress liver fibrosis. Here, we demonstrated that lentivirus-mediated knockdown of lnc-LFAR1 inhibits TGFβ-induced Smad2/3 phosphorylation and nuclear translocation. Furthermore, we also found that both the protein and mRNA levels of total Smad2/3 are also decreased in lnc-LFAR1 downregulated HSCs and HCs, whereas increased in lnc-LFAR1 up-regulated HSCs and HCs. These data suggest that lnc-LFAR1 promotes liver fibrogenesis by inducing Smad2/3 expression and phosphorylation, identifying a positive feedback loop in the TGFβ/Smad2/3/lnc-LFAR1 pathway. In addition, we also observed that lnc-LFAR1 promotes hepatic fibrosis by activating Notch signaling in vitro and in vivo, as demonstrated by an increased expression of the Notch receptors, Notch2 and Notch3, and Notch target gene Hes1. Further study revealed that lnc-LFAR1 associates with Smad2/3 and increases the binding of Smad2/3 to the promoters of Notch2 and Notch3. Although we have revealed that lnc-LFAR1 interacts with TGFβR1, by RIP, which subsequently phosphorylates Smad2/3 to promote its nuclear translocation and the binding to the target promoters as shown by ChIP, we will perform RIP-seq to further confirm the specificity of lnc-LFAR1-TGFβR1 interaction and ChIP-seq for the binding of Smad2/3 to specific sites. Collectively, the data reported in this study have demonstrated, for the first time, that a novel link connecting TGFβ with Notch pathway to regulate the progression of liver fibrosis.

In summary, our data may identify lnc-LFAR1/Smad2/3 nexus as a novel regulator of TGFβ and Notch pathways in liver fibrosis, suggesting that lnc-LFAR1 may be a candidate anti-fibrotic target.

## Methods

**Cell Culture**. The non-tumorigenic mouse HC cell line AML12 was maintained in Dulbecco's modified Eagle's medium (Invitrogen, Camarillo, CA, USA) supplemented with 10% fetal bovine serum, 1× insulin-transferrin-sodium selenite media supplement (ITS; Sigma-Aldrich), dexamethasone (40 ng ml$^{-1}$), penicillin (100 U ml$^{-1}$) and streptomycin (100 μg ml$^{-1}$). HEK293T cells were cultured in Dulbecco's modified Eagle's medium (Invitrogen) supplemented with 10% fetal bovine serum, penicillin (100 U ml$^{-1}$) and streptomycin (100 μg ml$^{-1}$). Both cells were cultured at 37 °C in an atmosphere containing 5% $CO_2$.

**Microarray and computational analysis**. Briefly, samples (five liver tissues from five Balb/c mice treated with CCl4 for 6 weeks and five liver tissues from five Balb/c mice treated with oil for 6 weeks) were used to synthesize double-stranded cDNA, and double-stranded cDNA was labeled and hybridized to an Affymetrix Mouse Gene ST 1.0 array (Affymetrix, Santa Clara, CA, USA), and after the washing steps the arrays were scanned using the GeneChip2 Scanner 3000 7 G (Affymetrix). Tiff images generated by the GeneChip2 Scanner 3000 7 G were processed by the Genepix 6microarray analysis software (Molecular Devices). For transcriptome assay once lnc-LFAR1 is knocked down in vivo, RNAs were collected from liver tissues of mice treated with oil in combination with injection of lenti-NC (NC, $n = 3$), CCl4 in combination with injection of lenti-NC (NC + CCl4, $n = 3$), oil in combination with injection of lenti-shLFAR1 (shLFAR1, $n = 3$) and CCl4 in combination with injection of lenti-shLFAR1 (shLFAR1 +CCl4, $n = 3$) and subjected to similar Affymetrix GeneChip® Mouse Genome 430 2.0 Array detection. Gene array analysis was performed according to the manufacturer's instructions, followed by data analysis as described. The threshold we used to screen up- or down-regulated lncRNAs and mRNAs was fold change >1.6 and $P < 0.05$. Microarray data have been deposited in NCBI Gene Expression Omnibus (GEO) under the following accession numbers GSE80601 (Affymetrix Mouse Gene ST 1.0 array) and GSE89147 (Affymetrix GeneChip® Mouse Genome 430 2.0 Array).

**RNA-Seq and computational analysis**. Briefly, primary HSCs infected with two separated lnc-LFAR1-shRNAs were collected and lysed with Trizol reagent. 1 μg RNA was used for library preparation with TruSeq Stranded Total RNA with Ribo-Zero Gold kit (Illumina). The sequenced reads were aligned to the mouse reference genome (NCBI37/mm9) using TopHat v1.4.1. Differential gene expression was performed with EdgeR (Empirical analysis of digital gene expression data in R) version 3.08. Adjusted $P$ values were computed using the Benjamini–Hochburg method. The threshold we used to screen up- or downregulated mRNAs was fold change >2 and padj <0.05. The transcriptome sequencing data have been deposited in NCBI Gene Expression Omnibus (GEO) under the following accession number: GSE96526.

**Animals in vivo study**. Animal protocols were approved by Tianjin Medical University Animal Care and Use Committee. The methods were carried out in accordance with the approved guidelines. All Balb/c and C57BL/6 J male mice aged

at 8 weeks obtained from *Institute of Laboratory Animal Sciences, CAMS & PUMC* (Beijing, China), weighting about 20 g. Mice were maintained in a 12-h light/dark cycle at 22–25 °C with free access to food and water. After acclimatization for one week, the hepatic fibrosis mice model was produced by three injections of carbon tetrachloride (CCl₄, Sigma-Aldrich, St. Louis, MO, USA) per week for 2–10 weeks or by BDL (3–21 days). For CCl₄-induced mouse liver fibrosis model, forty Balb/c mice were randomly divided into four groups: Mice were treated with oil in combination with injection of lenti-NC (NC, n = 10), CCl₄ in combination with injection of lenti-NC (NC+CCl₄, n = 10), oil in combination with injection of lenti-shLFAR1 (shLFAR1-3, n = 10) and CCl₄ in combination with injection of lenti-shLFAR1 (shLFAR1-3+CCl₄, n = 10). The lentivirus was injected via the tail vein 2 weeks after the first injection of CCl₄ (1 × 10⁹ pfu per mouse)[20]. Mice in the NC+CCl₄ group and the shLFAR1-3+CCl₄ group, separately, were administered 5% CCl₄ (v/v) dissolved in olive oil (0.3 ml kg⁻¹ body weight) thrice per week for additional 4 weeks via intraperitoneal (ip) injection after the lentivirus was injected. The shLFAR1-3 group animals were injected with an equivalent volume of olive oil. After treatment with CCl₄ for 6 weeks, all of mice were killed under anesthesia with 3% sodium pentobarbital (45 mg kg⁻¹, ip). For BDL-induced mouse liver fibrosis model, ninety Balb/c mice were randomly divided into six groups: Mice were treated with sham operation in combination with injection of lenti-NC (NC, n = 15), BDL operation in combination with injection of lenti-NC (NC+BDL, n = 15), sham operation in combination with injection of lenti-shLFAR1-1 (shLFAR1-1, n = 15), BDL operation in combination with injection of lenti-shLFAR1-1 (shLFAR1-1+BDL, n = 15), sham operation in combination with injection of lenti-shLFAR1-3 (shLFAR1-3, n = 15) and BDL operation in combination with injection of lenti-shLFAR1-1 (shLFAR1-3+BDL, n = 15). The lentivirus was injected via the tail vein 1 day before the surgical operation (1 × 10⁹ pfu per mouse)[20]. Twenty-one days later, all of mice were killed under anesthesia with 3% sodium pentobarbital (45 mg kg⁻¹, ip). Liver specimens and serums were obtained for analyses of liver functions, mRNA and protein expression of fibrotic indexes by real-time RT-PCR, western blot, histology and IHC.

**Isolation and culture of primary HSCs and HCs.** Primary mouse HSCs and HCs were isolated by pronase/collagenase perfusion digestion followed by subsequent density gradient centrifugation. In brief, male 40-week-old male Balb/c mice weighing 25–30 g used in the study received human cares. Liver was initially in situ digested with 0.05% pronase E (Roche, Mannheim, Germany), 0.03% collagenase type IV (Sigma-Aldrich, St. Louis, MO, USA) and then further digested with collagenase type IV, pronase E and DNase I (Roche) solution at 37 °C bath shaking for 20 min. Subsequently, HSCs were isolated from non-parenchymal cells using 8.2, 12 and 18% Nycodenz solution (Sigma-Aldrich) at 1450 g and 4 °C without brake for 22 min due to the feature of the massive amount of vitamin A-storing lipid droplets in them. In addition, the purity of the isolated population was tested by the characteristics of star-like shape, perinuclear lipid droplets and α-SMA staining. HCs were isolated from the 10-week-old male Balb/c mice weighting 20 g by in situ perfused with 30 ml SC1 solution and 30 ml 0.05% Collagenase IV solution sequentially. Then HCs were pelleted by centrifugation 50 g for 4 min three times. Cell viability was determined by the trypan blue exclusion method. Primary HSCs and HCs were cultured in high-glucose Dulbecco's modified Eagle's medium containing 10% fetal bovine serum (FBS) and 1% penicillin/streptomycin and maintained in a humidified incubator with 5% CO₂ at 37 °C.

**Histology and immunohistochemistry.** The specimen was sequentially fixed in 10% formalin for 2 days, transferred to ethanol of different concentration and embedded in paraffin in preparation for histopathological analysis. Thin sections (5 μm) were stained with H&E and Sirius red for histopathological study. According to the above results, three sections were chosen from each group for IHC analysis. Briefly, sections prepared on slides were first submitted to antigen retrieval by incubation in citrate buffer (pH 6.0) for 5 min at 108 °C and pretreated with 3% H₂O₂ in phosphate-buffered saline (PBS) for 15 min at room temperature followed by washing with PBS. Slides were subsequently incubated in normal goat serum for 20 min to block the nonspecific immunoreactivity. Next, the slides were treated with primary antibody α-SMA (1:50, rabbit polyclonal, Abcam, ab5694), collagen1 (1:1000, rabbit polyclonal, Abcam, ab34710), TGFβ (1:50, rabbit polyclonal, Abcam, ab66043), Hes1 (1:100, rabbit polyclonal, Abcam, ab71559) or Notch3 (1:250, rabbit polyclonal, Abcam, ab23426) overnight at 4 °C. In addition, tissue sections were processed omitting the primary antibody as the negative control. The slides were incubated with secondary antibody (1:500) (horseradish peroxidase-conjugated anti-rabbit IgG) and the reaction products were visualized using diaminobenzidine (DAB) and monitored by microscopy. Morphometrical analysis was performed for five random fields in each preparation, and average percentages of fibrotic area are plotted.

**TUNEL assay.** For TUNEL staining, we used an in situ cell detection kit (Roche) according to the manufacturer's protocol. After dewaxing and rehydration, we pretreatment tissue sections with 3% H₂O₂ and subsequent proteinase K permeation. Pretreatment with DNase I served as a positive control and TUNEL

reaction mixture lacking terminal transferase (TdT) as a negative control. Samples were analyzed by light microscopy.

**Hydroxyproline assay.** Total collagen content was tested by measuring the amount of hydroxyproline in liver tissue using commercially available hydroxyproline detection kits purchased from Nan Jing Jan Cheng Biochemical Institute (Nanjing, China) according to the manufacturer's instructions.

**Apoptosis assay.** The apoptosis of cells with different treatment were analyzed using a fluorescein isothiocyanate (FITC) Annexin V Apoptosis Detection Kit I (BD Biosciences) according to the manufacturer's instruction. Briefly, AML12 cells were transfected with siRNA targeting lnc-LFAR1 or siRNA-control for 48 h and then treated with or without 10 ng ml⁻¹ TGFβ (R&D) for 48 h; cells were scraped and washed twice with ice-cold PBS and then resuspended in 1× binding buffer at a concentration of 1 × 10⁶ cells per ml. Next 100 μl of the solution, 5 μl of FITC Annexin V and 5 μl PI were sequentially transferred to a 5 ml culture tube followed by gently vortex and incubating for 15 min at 25 °C in the dark. Finally 400 μl of 1× binding buffer was added to each tube and analyzed by flow cytometry within 1 h.

**Confocal microscopy.** Freshly isolated primary HSCs and AML12 cells were re-plated on Poly-lysine-pre-coated glass cover slips and incubated overnight at 37 °C to reach typical adhesion and spreading. Cells were transfected with siRNA targeting lnc-LFAR1 or siRNA-control for 48 h, recombinant TGFβ (R&D) was then added to cells transfected with siRNA-lnc-LFAR1 or siRNA-control for additional 24 h and then cells were sequentially fixed with 4% paraformaldehyde in PBS overnight at 4 °C, permeabilized with 1% Triton X-100 in PBS for 30 min and blocked using 5% bovine serum albumin (BSA) in tris-buffered saline-Tween 20 (TBST) with 0.1% Tween-20 for 30 min at room temperature. Next, the cells were incubated with primary antibodies against α-SMA (1:300, rabbit polyclonal, Abcam, ab5694), Col1α1 (1:500, rabbit polyclonal, Abcam, ab34710), Smad2/3 (1:125, rabbit polyclonal, Cell Signaling Technology, 5678s and rabbit monoclonal, Cell Signaling Technology, 8685), pSmad2/3 (1:200, rabbit monoclonal, Cell Signaling Technology, 8828) overnight at 4 °C and an irrelevant isotype rabbit IgG was used as a negative control. After washing in PBS, cells were incubated with FITC-conjugated secondary antibodies (1:100, Invitrogen) in PBS away from light for 1 h at room temperature. And the nuclei were stained with DAPI (5 μg ml⁻¹). Finally, the slides were washed with PBS and the cover slips were mounted with an anti-fade Mounting Medium (P0126, Beyotime, Shanghai, China). All immunofluorescence was then visualized by a confocal microscope (LSM 700). Every experiment was repeated at least three times independently.

**Quantitative real-time polymerase chain reaction.** Total RNA extracted from liver tissues or cells with Trizol reagent (Takara, Dalian, China), nuclear and cytoplasmic RNA prepared using PARIS™ Kit (Invitrogen) and the RNA isolated using the Magna RIP kit were measured with a NanoDrop ND-2000 spectrophotometer (Life Technologies, Grand Island, NY, USA). All RNA was digested with DNase I (Takara). Briefly, the 10 μl RT reactions (1 μg RNA, 1 μl buffer, 1 μl DNase1 and water) was incubated for 15 min at 37 °C followed by adding 1 μl of EDTA, incubated for 10 min at 65 °C and then maintained at 4 °C. Next, the first-strand cDNA was synthesized using AMV Reverse Transcriptase (Thermo Fisher Scientific, Basingstoke, UK) according to the manufacturer's instructions. For real-time PCR, all reactions were performed in triplicate with SYBR Green master mix (Takara) under the following conditions: 15 min at 95 °C for initial denaturation, followed by 40 cycles of segments of 95 °C for 30 s and 60 °C for 30 s in the Light Cycler®96 Real-Time PCR System (Roche). The expression levels of housekeeping gene β-actin were used to normalize the expression levels of the genes-of-interest. The sequences of primers for real-time PCR are listed in Supporting Table 3.

**Western blot and immunoprecipitation.** Cells were lysed with cell lysis buffer (Cell Signaling Technology) supplemented with protease inhibitor cocktail, 1% phenylmethanesulfonyl fluoride (PMSF) and 1% phosphatase inhibitor. Protein concentrations were measured by the BCATM Protein Assay Kit (Bio-Rad Laboratories, Hercules, CA, USA) using BSA as standard. Appropriate amount of protein samples along with 4× loading buffer and ddH₂O were boiled for 4 min and then subjected to sodium dodecyl sulphate–polyacrylamide gel electrophoresis. Following by electrophoresis, the separated proteins were blotted onto poly-vinylidene fluoride (PVDF) membranes in transfer buffer with constant current of 300 mA for 3 h at 4 °C. Then the PVDF membranes were sequentially washed with TBST containing 0.2% Tween-20, blocked with 5% nonfat milk in TBST and incubated with the interested primary antibodies of α-SMA (1:1000, rabbit poly-clonal, Abcam, ab5694), col1α1 (1:1000, rabbit polyclonal, Abcam, ab34710), TGF-β (1:2000, rabbit polyclonal, Abcam, ab66043), TGFβR1(1:1000, rabbit polyclonal, Abcam, ab31013), MMP2 (1:2000, rabbit monoclonal, Abcam, ab92536), Notch2 (1:1000, rabbit monoclonal, Cell Signaling Technology, 5732), Notch3 (1:1000, rabbit polyclonal, Abcam, ab23426), Hes1 (1:1000, rabbit polyclonal, Abcam, ab71559), Smad2/3 (1:1000, rabbit polyclonal, Cell Signaling Technology, 5678 s and rabbit monoclonal, Cell Signaling Technology, 8685) and pSmad2/3 (1:1000, rabbit monoclonal, Cell Signaling Technology, 8828) diluted in TBST containing

0.2% Tween-20 overnight at 4 ℃. The levels of GAPDH were severed as control for total protein amount. Next, the membranes were incubated with secondary anti-body for 1 h at RT with shaking. Signal was detected using the chemiluminescence (ECL) system (Merck Millipore, Darmstadt, Germany). Every experiment was repeated at least three times independently. All the uncropped scans of the western blots are shown in Supplementary Figs 18 and 19.

**Nuclear-cytoplasmic fractionation**. Cytoplasmic and nuclear RNA Isolation were performed with PARIS™ Kit (Invitrogen) following the manufacturer's instruction. Briefly, the cells (AML-12, primary HSCs, primary HCs) were digested to individual cells with trypsin and the trypsin was inactivated with complete medium followed by centrifugation at 1100 r.p.m. for 3 min. The collected cells were washed twice with ice-cold PBS and lysed with 500 μl ice-cold cell fractionation buffer. Cells were gently re-suspend by vortex or pipetting and incubated on ice for 10 min. Centrifuge samples 3 min at 500 g to separates the nuclear and cytoplasmic cell fractions. The supernatant was transferred to a fresh RNase-free tube. In addition, the remaining lysate was washed with ice-cold cell fractionation buffer and centrifuged at 500 g for 1 min. Add 450 μl of ice-cold cell disruption buffer to lyse the nuclei until the lysate is homogenous. Mix the lysate and the supernatant above with a 2× lysis/binding solution and add equal volume of 100% ethanol to the mixture followed by drawing the mixture through a filter cartridge. Wash the sample sequentially with 700 μl wash solution, 1.5 ml wash solution 2/3. The RNA of cytoplasmic and nuclear was eluted with 40–60 μl of 95 ℃ elution solution. All fractionation steps were performed at 4 ℃ or on ice.

**5' and 3' rapid amplification of cDNA ends (RACE)**. We used the 5'-RACE and 3'-RACE analyses to determine the transcriptional initiation and termination sites of lnc-LFAR1 using a SMARTer™ RACE cDNA Amplification Kit (Clontech, Palo Alto, CA, USA) according to the manufacturer's instructions. In brief, RNA was isolated from mouse liver and 3'- and 5'-RACE-Ready cDNA were synthesized using SMARTScribe Reverse Transcriptase. Amplification was performed as follows: ● five cycles: 94 ℃ 30 s 72 ℃ 3 min; ● five cycles: 94 ℃ 30 s 70 ℃ 30 s 72 ℃ 3 min; ● 25 cycles 94 ℃ 30 s 68 ℃ 30 s 72 ℃ 3 min. The obtained band was gel purified and cloning with the linearized pRACE vector. The obtained band was then sequenced. The gene-specific primers used for the PCR of the RACE analysis are provided in supporting Table 3.

**Chromatin immunoprecipitation (ChIP)**. Briefly, primary HSCs and AML12 cells infected with LV-control or LV-lnc-LFAR1 for 72 h, and then treated with or without TGFβ for 24 h. Cells were seeded in cell cultures of 15 cm. Chromatin was cross-linked with 15 ml pre-heated 10% FBS media/1% formaldehyde for 10 min at room temperature (RT). Stop the fixation by addition of glycine to final concentration of 0.125 M and incubate for 5 min at RT. The cells were washed twice with 1× PBS and harvested in 670 μl SDS buffer containing protease inhibitors (PMSF). And then the fixed cells can be frozen at −80 ℃. Lysate was thawed and centrifuged at 1200 rpm followed by re-suspending with ice-cold IP Buffer (one volume SDS buffer and 0.5 volume Triton Dilution Buffer). Samples were sheared by sonication with a 5 s/15 s cycle at power setting 30% for 40 times and centrifuged at 20,000 g for 30 min. Transfer supernatant to new tubes and quantify the protein content from each sample by the BCA™ Protein Assay Kit (Bio-Rad Laboratories) using BSA as standard. Next, dilute samples with IP Buffer to a desired concentration and remove 10 μl (1%) of lysate used as total control. Each single IP was performed in 1 ml. Lysates incubated with primary antibodies (5 μg Smad2/3 or 5 μg IgG) overnight at 4 ℃ rotating. Add 50 μl beads by a wide-bore pipet tip and incubate on a rotating wheel 4 h at 4 ℃. Next, immunoprecipitated complexes were sequentially washed two times with wash-buffer (1 ml 150 mM wash buffer and 500 mM wash buffer) followed by adding 120 μl of 1% SDS, 0.1 M NaHCO3 and incubating overnight at 65 ℃. Finally, DNA was purified with PCR purification kit (Qiagen) and used as templates for PCR reactions. Primers used for PCR in ChIP experiments are described in Supporting Table 3.

**RNA immunoprecipitation (RIP)**. RIP experiments were performed using the Magna RIP RNA-Binding Protein Immunoprecipitation Kit (Millipore, Bedford, MA, USA) according to the manufacturer's instructions. Briefly, the single-cell suspensions isolated from mouse liver or the AML12 cells at ~ 90% confluency in culture dishes (15 cm) were sequentially washed twice with ice-cold PBS, harvested into 15 ml conical tubes with 10 ml ice-cold PBS and collected by centrifugation at 1500 r.p.m. for 5 min at 4 ℃. Next, resuspended the cell pellet in an equal pellet volume of complete RIP Lysis Buffer and incubate the lysate on ice for 5 min followed by storing at −80 ℃. Thaw the RIP lysate quickly and centrifuge at 14,000 r.p.m. for 10 min at 4 ℃. Transfer 10 μl supernatant to new tubes as input and another 100 μl supernatant to the beads-antibody complex followed by adding 900 μl RIP immunoprecipitation buffer for each RIP. The antibodies used for RIP were 5 μg Smad2/3 (rabbit polyclonal, Cell Signaling Technology, 5678 s and rabbit monoclonal, Cell Signaling Technology, 8685), 5 μg SUZ12 (rabbit polyclonal, Abcam, ab12073), 5 μg Ago2 (rabbit polyclonal, Abcam, ab32381), 5 μg TGFβR1 (rabbit polyclonal, Abcam, ab31013) and 5 μg IgG (Millipore, PP64B). And all

the tubes were incubated with rotating overnight at 4 ℃. Centrifuge the immunoprecipitation tubes briefly and discard the supernatant using a magnetic separator. Then wash the beads and purification the RNA. The precipitated RNAs were detected by qRT-PCR. The gene-specific primers used for detecting lncRNA-LFAR1 are presented in supporting Table 3.

**Plasmid constructs and luciferase activity assays**. To construct the EGFP tag expression vectors. The transcription potential of lnc-LFAR1 was measured in CPC (http://cpc.cbi.pku.edu.cn/) and the predicted ORFs of lnc-LFAR1 (154 bp, 31 bp before predicted ORF start codon and without stop codon), a annotated lnc-MALAT1 (440 bp, 244 bp before predicted ORF start codon and without stop codon), and the first exon sequence of a protein-coding gene GAPDH (263 bp, 16 bp before ATG start codon and without stop codon) were amplified by using the primers depicted in Supporting Table 3. These ORFs were cloned into pcDNA3.1(+) between HindIII and BamHI sites, with C-terminal EGFP between BamHI and XhoI. These vectors were transfected into AML12 cells separately with Lipofectamine 2000 (Invitrogen), according to the manufacturer's instructions. After 48 h of transfection, cells transfected with EGFP plasmids were visualized by fluorescent microscopy. To construct the luciferase reporter plasmids, the 5'-untranlated regions spanning from +1 to −2000 relative to the transcription start site of the predicted target gene LFAR1 containing potential Smad2/3 binding sites were amplified by PCR from mouse genomic DNA. The PCR products were digested with KpnI and XhoI and cloned into the promoterless pGL3-Basic vector (Promega, Madison, WI, USA) (pGL3-LFAR1) immediately downstream of the luciferase gene. The QuikChange Lightning Site-Directed Mutagenesis Kit (Agilent Technologies, Santa Clara, CA, USA) was used for mutation of Smad2/3 binding sites. All the construct sequences were verified by automated DNA sequencing. For experimental validation of selected Smad2/3 targets, AML12 cells were transfected with pGL3- LFAR-1 constructs containing wild-type (wt) or mutant (mut) Smad2/3 expression vectors. Each sample was co-transfected with the pRL-TK vector as an internal control for transfection efficiency. After 48 h transfection, cells were incubated with TGFβ for 24 h. Luciferase assays were performed using the dual-luciferase reporter assay system (Promega, Madison, WI, USA). The relative firefly luciferase activity was normalized with renilla luciferase activity. PCR primers are listed in Supporting Table 3.

**Lentivirus production and construction**. Oligos encoding shRNA specific for lnc-LFAR1, Smad2 and Smad3 were ligated into pSUPER.retro.puro, and the fragment containing the H1 promoter and hairpin sequences was subcloned into the lentiviral shuttle pCCL.PPT.hPGK.GFP.Wpre (a kind gift from Dr. ZhenyiMa at Tianjin Medical University). The full-length lnc-LFAR1 cDNA was sequentially amplified by PCR and ligated into the lentiviral shuttle pCCL.PPT.hPGK.IRES.eGFP/preto generate the over-expression plasmid. These plasmids were used to produce lentivirus in HEK-293T cells with the packaging plasmids pMD2.BSBG, pMDLg/pRRE and pRSV-REV. Infectious lentiviruses were harvested at 36 and 60 h after transfection and filtered through 0.45 μm PVDF filters. Recombinant lentiviruses used in vivo were concentrated 100-fold by ultracentrifugation (2 h at 120,000 g). The virus-containing pellet was dissolved in PBS and injected in mice within 48 h. The primer sets used are shown in Supporting Table 3.

**RNA interference**. For gene knockdown analysis, small interferring (si) RNA targeting the lnc-LFAR1 and TGFβR1 sequences and non-targeting siRNA were obtained from GenePharma Biological Technology (Shanghai, China). Cells were transfected with the siRNAs at 50% confluence using lipofectamine MAX according to the manufacturer's instructions (Invitrogen). After culturing for 48 h, cells were analyzed by real-time PCR to determine knockdown efficiency. Target sequences of these siRNA are listed in supporting Table 3.

**Statistical analysis**. Data were expressed as mean ± s.e.m. All the statistical analysis was performed with the SPSS 13.0 (IBM, Armonk, NY, USA). Statistical analysis was performed using either Student's t-test (two-group comparison) or one-way analysis of variance (more than two groups) followed by post hoc comparison, and differences with $P < 0.05$ were considered significant.

**Data availability**. Microarray and RNA-seq data that support the findings of this study have been deposited in GEO under the following accession numbers GSE80601, GSE89147 and GSE96526. The authors declare that all other data supporting the findings of this study are available within the article and its Supplementary Information files, or from the corresponding authors on request.

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

## Acknowledgements

This work was supported by the National Natural Science Foundation of China (No. 81670558 to W.H.), and the Ministry of Science and Technology (No. 2016ZX10002008-007 to T.H.), and Tianjin Municipal Science and Technology Commission (No. 13RCGFSY19200 to T.H.), and the 'high-level innovation talent' grant (116001-20100097 to W.H.).

## Author contributions

K.Z. and W.H. conceived and designed the studies. K.Z. and X.H. performed the majority of the experiments. Y.C. and Z.S. performed the confocal microscopy assay. Z.H. and H.C. performed histological and immunocytochemistry analyses. Y.H., Z.Z. and T.C. performed plasmid constructs and luciferase activity assays. Q.Y., G.S., C.L., M.S., L.Z. and Z.Y. gave technical support and conceptual advice. K.Z., T.H. and W.H. analyzed the data and wrote the manuscript. All authors critically reviewed and approved the final manuscript.

## Additional information

**Competing interests:** The authors declare no competing financial interests.

