## [Peer review file · Nature Communications]

Reviewers' comments:

Reviewer #1 (Remarks to the Author):

This is an interesting manuscript that highlights the novel functional role of a lncRNA (lnc-LFAR1) in driving hepatic stellate cell fibrogenesis and hepatocyte apoptosis, and provide a mechanistic link from this lncRNA to SMAD2/3 and Notch signaling.

1. The authors state that lnc-LFAR1 is liver-enriched but do not show data about its relative expression in other tissues. A tissue survey of its expression in normal tissues should be included, acknowledging that its levels could change with disease in other tissues apart from liver.

2. The in vivo studies need to be expanded. A second model of liver injury in which lnc-LFAR1 is antagonized is required to exclude the possibility that lnc-LFAR1 alters the metabolism or toxicity of CCl₄ rather than by altering stellate cell responses. Moreover, since lnc-LFAR1 is also expressed by hepatocytes, it is not clear whether the effects of knockdown in vivo are attributable to altered biology of hepatocytes, stellate cells or both. Also, TUNEL staining should be performed to quantify apoptosis in situ, especially in hepatocytes.

3. Data on the extent of liver injury by assessment of serum AST, ALT and histologic scoring of inflammation and fibrosis should be included in in vivo experiments.

4. It would be highly informative to perform transcriptome analysis of in vivo studies by either RNAseq or microarray with pathway analysis to identify pathways affected by lnc-LFAR1 knockdown in whole liver. This would yield a clearer indication of its targets and mechanisms of action in vivo. Also, in Figure 4, when was sh lnc-LFAR1 administered relative to the start of CCl₄ intoxication?

5. Figure 1 shows differential expression of 'ten representative lncRNAs'. How were these ten chosen and what was the nature of other differentially expressed lncRNAs?

6. Figures 2a and 2b, show cellular localization and cell-specific levels of lncLFAR1 in normal liver only. Do these ratios change in fibrotic vs. normal liver? If the nuclear fraction increases in fibrosis (or with tgf stimulation) this would support the finding that the RNA binds with smad2/3 and promotes binding to target promoters. Similarly, given the unexpected finding that lncLFAR1 is downregulated in fibrosis in whole liver, but upregulated in HSCs in activation, it is important to show that lncLFAR1 is increased in HSCs in fibrotic livers either by in situ methods or quantification in freshly isolated HSCs from normal and injured liver.

7. There are countless spelling errors, syntactical problems, inconsistencies in terminology and organizational problems throughout the manuscript, too numerous to list here (e.g., binding instead of binding, cultrue instead of culture'). This work should be meticulously reviewed by a native English speaker and by a spelling check program. Also In vivo data from the same experiments are scattered in several figures (fig 4, fig 6, fig7) and in vivo and in vitro data are mixed within the same figures (fig 6, fig 7). Also, the figure 3a's y-axis is labeled "mRNA", but since lncLFAR1 is one of the independent variables, it should be labeled "RNA".

8. Quantification of Sirius red, aSMA, collagen1a1 is necessary (fig4). Also, a better Western is required as no band is detected in control lane (i.e. Fig 4b).

9. In supplementary fig 6. the authors show expression of collagen1a1 and aSMA in hepatocytes by qPCR, WB, and IF which is unexpected and raises concerns about the specificity of this staining. Similarly AML12 cells are a hepatocyte line so it is puzzling that they express collagen (Supplemental Figure 5) and profibrogenic genes.

10. The methods section states that differential expression in the microarray experiment was defined as 1.6 fold change and $p < 0.05$. Is this FDR rather than unadjusted p -value?

11. The authors should cite and comment upon a prior report characterizing lncRNAs in stellate cells and explore whether nc-LFAR1 was previously identified by others (Zhou, C., et al., Long noncoding RNAs expressed in human hepatic stellate cells form networks with extracellular matrix proteins. *Genome Medicine*, 2016. 8.)

Reviewer #2 (Remarks to the Author):

Zhang et al. identifies a novel lncRNA, lnc-LFAR1, that is specifically enriched in the liver. The paper proposes a positive feedback loop between TGF β signaling pathway and lnc-LFAR1 expression that regulates the progression of liver fibrosis. The effects of lnc-LFAR1 knockdown on gene and protein expression levels of pro-fibrotic, pro-inflammatory, and pro-apoptotic genes is shown mainly through qPCRs and western blots, respectively. lnc-LFAR1 knockdown is further reproduced in vivo, and is shown to have a phenotypic impact in fibrotic livers when analyzed by histology and immunohistochemistry. If Zhang and colleagues are able to produce similar knockdown results using different shRNAs to eliminate the possibility of off-target effects, then we could conclude that the lnc-LFAR1 has a very interesting regulatory role in liver fibrosis. Unfortunately, the paper does not provide any clear functional mechanism for lnc-LFAR1 beyond its involvement in the TGF β /Notch signaling pathway.

lnc-LFAR1 does seem to have a regulatory role in liver fibrosis, shown mainly through qPCRs and western blots. The Major Questions:

1. How many lncRNAs are enriched in the liver, but not other tissues? If lnc-LFAR1 is not the only lncRNA specifically enriched in the liver, what evidence suggested further investigation of lnc-LFAR1 rather than other potential candidates? Additionally, is the search for liver centric lncRNA expression only within the subgroup of lncRNAs that show differential expression with CCL4 treatment in mice? Please clarify the methodology from which lnc-LFAR1 is selected as the target.
2. Supplementary table provides 3 lnc-LFAR1-shRNAs, though Figure 3 only shows data for lnc-LFAR1-shRNA-3. Please also provide the data for the effects of other two shRNAs on expression of extracellular matrix genes to be sure that there are no off target effects. Please clarify which shRNA was used to silence lnc-LFAR1 in vivo. Please clarify which shRNA was used to show knockdown data of Smad2/3, and also provide results from the other shRNA knockdown.
3. The effects of lnc-LFAR1 on extracellular matrix gene expression will be more convincing with microarray or RNA-seq, preferentially using at least 2 separate shRNAs.
4. Will fully activated HSCs with high expression of lnc-LFAR1 at day 14 still exhibit knockdown of post-fibrogenic genes with lnc-LFAR1-shRNA? Moreover, will there still be minimal TGF β 1 stimulation of post-fibrogenic genes when lnc-LFAR1 is knocked down at later stage of HSCs? Once the fibrotic program gets activated, can it still be reversed, downregulated through regulation of lnc-LFAR1?
5. Supplementary Figure 5-8 illustrate that knockdown of lnc-LFAR1 reduces TGF β -induced pro-apoptosis gene expression and apoptosis in HCs, suggesting that low levels of lnc-LFAR1 in HCs is crucial for deterring HSCs activation. Though, lnc-LFAR1 is already downregulated in primary HCs isolated from mice treated with CCL4 according to Supplementary Figure 4a. How do the results described in Supplementary Figure 5-8 explain the phenomenon in Supplementary Figure 4?
6. The paper presents the idea that TGF β induces lnc-LFAR1 expression through Smad2/3 binding at the promoter region of lnc-LFAR1, which in turn positively feeds back into TGF β activation via Smad2/3 phosphorylation and thus progression of liver fibrosis. Figure 6 illustrates upregulation of pSmad2/3 with increased lnc-LFAR1 expression, and decreased pSmad2/3 with lnc-LFAR1 knockdown. Please show and explain how lnc-LFAR1 is having an effect on both total Smad2/3 levels as well as pSmad2/3 independent of total Smad2/3 levels. Data here is required.

Otherwise, we could conclude that lnc-LFAR1 has an indirect role rather than represent a direct functional mechanism.

Minor Errors and Clarifications:

1. Please clarify how the clusters were defined in the microarray in Figure 1d and Supplementary Figure 1b, and how the clusters were compared between normal and fibrotic livers. The manuscript states that 266 lncRNAs and 1007 mRNAs are upregulated, and 447 lncRNAs and 519 mRNAs are downregulated in the fibrotic livers. From our interpretation of the microarray, there are more lncRNAs that are upregulated than downregulated and more mRNAs that are downregulated than upregulated in CCL4 mice, which is the opposite trend of what is written.
2. Given that lnc-LFAR1 is present on the list of 10 randomly selected lncRNAs from hundreds of differentially regulated lncRNAs found in fibrotic liver, the selection process to validate microarray analysis does not seem random. Please clarify how these 10 lncRNAs are selected.
3. Please show relative level of mRNA for pro-fibrotic, pro-inflammation, and pro-apoptotic genes normalized to GAPDH in primary HSCs and primary HCs.
4. Were the C57 mice also randomly divided into 4 groups and received the same course of treatment as Balb/c mice?
5. Only 7 representative mRNAs are shown in Supplementary Figure 1c and 9 in Supplementary Figure 1d.
6. Is there significant difference of lnc-LFAR1 between HSCs and Kupffer cells in Figure 2b?
7. Are the primary HSCs cultured at day 12 culture-activated prior to TGFb1 stimulation and exhibit increase in level of aSMA and lnc-LFAR1 as shown in Figure 2d? If so, then is the increase in relative level of mRNA from TGFb1 stimulation a fold increase on top of culture-induced activation in Figure 2f?
8. Why is the level of TGFb1 lower in both LV-control and LV-lnc-LFAR1 in comparison to that in shRNA-control in Figure 3?
9. Misspelled "binding" in line 263.

Summary: The linkage of this lncRNA, LFAR1, to inflammatory genes and the fibrotic program is potentially of general interest and significance, but a more robust data set for mechanistic insight is required. Absent this, the paper falls short of the level of Nature Communications.

Response to Reviewers

Reviewers' comments:

Reviewer #1 (Remarks to the Author):

This is an interesting manuscript that highlights the novel functional role of a lncRNA (lnc-LFAR1) in driving hepatic stellate cell fibrogenesis and hepatocyte apoptosis, and provide a mechanistic link from this lncRNA to SMAD2/3 and Notch signaling.

1. The authors state that lnc-LFAR1 is liver-enriched but do not show data about its relative expression in other tissues. A tissue survey of its expression in normal tissues should be included, acknowledging that its levels could change with disease in other tissues apart from liver.

Answer 1: Thank you for your nice suggestion. We have performed a tissue survey of lnc-LFAR1 expression in both normal and liver-fibrosis mice, and the results showed that lnc-LFAR1 is liver-enriched compared with other tissues and its level changes only in liver during the liver fibrosis, as shown in Supplementary Fig. 2a.

2. The in vivo studies need to be expanded. A second model of liver injury in which lnc-LFAR1 is antagonized is required to exclude the possibility that lnc-LFAR1 alters the metabolism or toxicity of CCl₄ rather than by altering stellate cell responses. Moreover, since lnc-LFAR1 is also expressed by hepatocytes, it is not clear whether the effects of knockdown in vivo are attributable to altered biology of hepatocytes, stellate cells or both. Also, TUNEL staining should be performed to quantify apoptosis in situ, especially in hepatocytes.

Answer 2: Thank you for your nice suggestion. To exclude the possibility that lnc-LFAR1 alters the metabolism or toxicity of CCl₄ rather than by altering stellate cell responses, we have made a bile duct ligation (BDL)-induced liver fibrosis mice model, with which we have performed experiments done in CCl₄-induced mice model and obtained the same conclusion. The results were shown in Supplementary Fig. 10 and Supplementary Fig. 11. Moreover, we have also isolated both the primary HSCs and HCs from mice of the two models to investigate the biology of HSCs and HCs, and the results showed that lnc-LFAR1 silencing inhibits liver fibrosis through repressing the activation of HSCs and the apoptosis of hepatocytes (Supplementary Fig. 9 and Supplementary Fig. 11). In addition, TUNEL staining was

also performed to quantify apoptosis *in situ*, especially in hepatocytes (Supplementary Fig. 9 and Supplementary Fig. 10).

3. Data on the extent of liver injury by assessment of serum AST, ALT and histologic scoring of inflammation and fibrosis should be included in *in vivo* experiments.

Answer 3: Thank you for your nice suggestion. We have assessed serum AST, ALT (Supplementary Table. 1 and Supplementary Table. 2) and underwent histologic scoring of HE, Sirius Red and IHC by Quantity One software, which was performed for five random fields in each preparation, and average percentages of fibrotic area are plotted. The data are shown in relevant figures. Experiments were performed with the CCl₄- and BDL-induced mice models.

4. It would be highly informative to perform transcriptome analysis of *in vivo* studies by either RNAseq or microarray with pathway analysis to identify pathways affected by lnc-LFAR1 knockdown in whole liver. This would yield a clearer indication of its targets and mechanisms of action *in vivo*. Also, in Figure 4, when was shlnc-LFAR1 administered relative to the start of CCl₄ intoxication?

Answer 4: Thank you for your nice suggestion. We have performed mRNA microarrays of whole liver to analyze the effect of lnc-LFAR1 knockdown on CCl₄-induced liver fibrosis as shown in Fig. 4. The data revealed that 1598 mRNAs were up-regulated and 620 mRNAs were down-regulated in the CCl₄-treated mice infected with lenti-NC, while only 292 mRNAs were up-regulated and 112 mRNAs were down-regulated in the CCl₄-treated mice infected with lenti-shLFAR1. Moreover, there are 133 mRNAs were up-regulated and 178 mRNAs were down-regulated in the CCl₄-treated mice infected with lenti-shLFAR1, compared with the CCl₄-treated mice infected with lenti-NC (Fig. 4b). Among these, 234 out of 311 genes were in common with the set of genes deregulated in the CCl₄-treated mice infected with lenti-NC (Fig. 4c). Additionally, the GO and KEGG pathway analysis revealed that lnc-LFAR1 silencing affects a list of genes associated with collagen fibril organization and TGFβ receptor signaling pathway (Fig. 4d,e and Supplementary Fig. 9c,d). These results yield a clearer indication of its targets and mechanisms of action *in vivo*. In Figure 4, lenti-shlnc-LFAR1 was intravenously injected into CCl₄-treated mice via the tail vein 2 weeks after the first injection of CCl₄.

5. Figure 1 shows differential expression of 'ten representative lncRNAs'. How were these ten chosen

and what was the nature of other differentially expressed lncRNAs?

Answer 5: Thank you for your nice suggestion. Ten lncRNAs were chosen according to the fold change and the lncRNA-mRNA co-expression network. Two of them are liver specific. One is lnc-LFAR1, which is highly expressed in liver and HSCs. Since HSCs are the most principal cellular players promoting synthesis and deposition of ECM proteins in liver fibrosis, we choose lnc-LFAR1 for further investigation. The other one is not only liver enriched, but also hepatocyte specific, and we are also conducting researches on its function.

6. Figures 2a and 2b, show cellular localization and cell-specific levels of lncLFAR1 in normal liver only. Do these ratios change in fibrotic vs. normal liver? If the nuclear fraction increases in fibrosis (or with tgf stimulation) this would support the finding that the RNA binds with smad2/3 and promotes binding to target promoters. Similarly, given the unexpected finding that lncLFAR1 is downregulated in fibrosis in whole liver, but upregulated in HSCs in activation, it is important to show that lncLFAR1 is increased in HSCs in fibrotic livers either by in situ methods or quantification in freshly isolated HSCs from normal and injured liver.

Answer 6: Thank you for your nice suggestion. We isolated primary HSCs from normal and fibrotic mice, with which the nucleus and cytoplasm fractions were extracted to detect subcellular localization of lnc-LFAR1. The results showed that the ratio was not changed (Fig. 2a). Although the ratio was not changed, the total amount of lnc-LFAR1, including the nuclear and cytoplasmic lnc-LFAR1, was increased in fibrotic HSCs and the nuclear localized lnc-LFAR1 binds with Smad2/3 and promotes binding to target promoters. Furthermore, the similar results were obtained from AML12 cells treated with TGF β (Supplementary Fig. 2b). We have detected lnc-LFAR1 expression in freshly isolated HSCs from normal and injured liver. As shown in Fig.2c, lncLFAR1 is increased in HSCs in fibrotic livers.

7. There are countless spelling errors, syntactical problems, inconsistencies in terminology and organizational problems throughout the manuscript, too numerous to list here (e.g., binding instead of binding, cultrue instead of culture'). This work should be meticulously reviewed by a native English speaker and by a spelling check program. Also In vivo data from the same experiments are scattered in several figures (fig 4, fig 6, fig7) and in vivo and in vitro data are mixed within the same figures (fig 6, fig 7). Also, the figure 3a's y-axis is labeled "mRNA", but since lncLFAR1 is one of the independent

variables, it should be labeled "RNA".

Answer 7: Thank you for your criticisms. We have carefully checked the manuscript and corrected all the errors and problems. For example: "bingding" was corrected as "binding"; "phosphrylation" was corrected as "phosphorylation"; "cultrue" was corrected as "culture"; "purificate" was corrected as "purification". For better understanding, we have separated *in vivo* and *in vitro* data, originally organized in Fig 6, to Fig. 6 and Supplementary Fig. 14, respectively. In some of figures, *in vivo* and *in vitro* data are mixed in order to clearly illustrate one question. In addition, the label of y-axis in Fig. 3a has been corrected to "RNA".

8. Quantification of Sirius red, α SMA, collagen1 α 1 is necessary (fig4). Also, a better Western is required as no band is detected in control lane (i.e. Fig 4b).

Answer 8: Thank you for your suggestion. We have quantified the Sirius red, α -SMA, collagen1 α 1 and refined western blot data in Fig. 4b.

9. In supplementary fig 6. the authors show expression of collagen1 α 1 and α SMA in hepatocytes by qPCR, WB, and IF which is unexpected and raises concerns about the specificity of this staining. Similarly AML12 cells are an hepatocyte line so it is puzzling that they express collagen (Supplemental Figure 5) and profibrogenic genes.

Answer 9: Thank you for your suggestion. In supplementary Fig. 6, collagen1 α 1 and α -SMA were detected by qPCR and WB, however, the protein level of α -SMA is quite low in hepatocytes which can be visualized by long time exposure of WB. IF showed collagen1 α 1 and TGF β are expressed. Our data showed that hepatocytes express collagen and profibrogenic genes, which are consistent with previous reports ("Tu X, Zhang H, et al. MicroRNA-101 suppresses liver fibrosis by targeting the TGF β signalling pathway. The Journal of pathology 2014; 234:46-59"; "Park JH, Lee MK, Yoon J. Gamma-linolenic acid inhibits hepatic PAI-1 expression by inhibiting p38 MAPK-dependent activator protein and mitochondria-mediated apoptosis pathway. Apoptosis. 2015 Mar; 20(3):336-47" and "Rezvani M, Español-Suñer R, et al. In Vivo Hepatic Reprogramming of Myofibroblasts with AAV Vectors as a Therapeutic Strategy for Liver Fibrosis. Cell Stem Cell. 2016 Jun 2; 18(6):809-16").

10. The methods section states that differential expression in the microarray experiment was defined as

1.6 fold change and $p < 0.05$. Is this FDR rather than unadjusted p-value?

Answer 10: Thank you for your question. We have consulted the company which performed the microarray experiment and was informed that the p-value is not FDR.

11. The authors should cite and comment upon a prior report characterizing lncRNAs in stellate cells and explore whether nc-LFAR1 was previously identified by others (Zhou, C., et al., Long noncoding RNAs expressed in human hepatic stellate cells form networks with extracellular matrix proteins. *Genome Medicine*, 2016. 8.)

Answer 11: Thank you for your suggestion. We have carefully read that paper which shows a good work. It is related with liver fibrosis and provided us important inspirations for future study. We have cited and commented this report in the revised manuscript. We found that lnc-LFAR1 was not previously identified by others including the paper by Zhou. Thanks again for the helpful suggestion.

Reviewer #2 (Remarks to the Author):

Zhang et al. identifies a novel lncRNA, lnc-LFAR1, that is specifically enriched in the liver. The paper proposes a positive feedback loop between TGF β signaling pathway and lnc-LFAR1 expression that regulates the progression of liver fibrosis. The effects of lnc-LFAR1 knockdown on gene and protein expression levels of pro-fibrotic, pro-inflammatory, and pro-apoptotic genes is shown mainly through qPCRs and western blots, respectively. lnc-LFAR1 knockdown is further reproduced in vivo, and is shown to have a phenotypic impact in fibrotic livers when analyzed by histology and immunohistochemistry. If Zhang and colleagues are able to produce similar knockdown results using different shRNAs to eliminate the possibility of off-target effects, then we could conclude that the lnc-LFAR1 has a very interesting regulatory role in liver fibrosis. Unfortunately, the paper does not provide any clear functional mechanism for lnc-LFAR1 beyond its involvement in the TGF β /Notch signaling pathway.

lnc-LFAR1 does seem to have a regulatory role in liver fibrosis, shown mainly through qPCRs and

western blots. The Major Questions:

1. How many lncRNAs are enriched in the liver, but not other tissues? If lnc-LFAR1 is not the only lncRNA specifically enriched in the liver, what evidence suggested further investigation of lnc-LFAR1 rather than other potential candidates? Additionally, is the search for liver centric lncRNA expression only within the subgroup of lncRNAs that show differential expression with CCL₄ treatment in mice? Please clarify the methodology from which lnc-LFAR1 is selected as the target.

Answer 1: Thank you for your nice suggestion. Ten lncRNAs were chosen according to the fold change and the lncRNA-mRNA co-expression network. Two of them are liver specific. One is lnc-LFAR1, which is highly expressed in liver and HSCs. Since HSCs are the most principal cellular players promoting synthesis and deposition of ECM proteins in liver fibrosis, we choose lnc-LFAR1 for further investigation. The other one is not only liver enriched, but also hepatocyte specific, and we are also conducting researches on its function. In this study, we search for liver centric lncRNA expression within the subgroup of lncRNAs that show differential expression with CCL₄ treatment in mice.

2. Supplementary table provides 3 lnc-LFAR1-shRNAs, though Figure 3 only shows data for lnc-LFAR1-shRNA-3. Please also provide the data for the effects of other two shRNAs on expression of extracellular matrix genes to be sure that there are no off target effects. Please clarify which shRNA was used to silence lnc-LFAR1 *in vivo*. Please clarify which shRNA was used to show knockdown data of Smad2/3, and also provide results from the other shRNA knockdown.

Answer 2: Thank you for your nice suggestion. We designed 3 lnc-LFAR1-shRNAs and tested the knockdown effect of all lnc-LFAR1-shRNAs and found that lnc-LFAR1-shRNA-1 and 3 were more effective. So we detected the effects of lnc-LFAR1-shRNA-1 and 3 on expression of extracellular matrix genes, as shown in Fig. 3, confirming that there are no off target effects.

We have clarified that lnc-LFAR1-shRNA-3 was used in CCL₄-induced mice liver fibrosis model. In addition, as suggested by the editor and another reviewer, we have made a bile duct ligation (BDL)-induced liver fibrosis mice model to exclude the possibility that lnc-LFAR1 alters the metabolism or toxicity of CCL₄ rather than by altering stellate cell responses, and used both lnc-LFAR1-shRNA-1 and lnc-LFAR1-shRNA-3 in BDL-induced liver fibrosis model. The results are shown in Supplementary Fig. 10 and Supplementary Fig. 11.

We have used two Smad2/3 shRNAs to re-do the experiment of knockdown of Smad2/3. The results are shown in Fig. 5 and Supplementary Fig. 12a.

3. The effects of lnc-LFAR1 on extracellular matrix gene expression will be more convincing with microarray or RNA-seq, preferentially using at least 2 separate shRNAs.

Answer 3: Thank you for your nice suggestion. The other reviewer also suggested us to perform microarray with pathway analysis to identify pathways affected by lnc-LFAR1 knockdown in whole liver. Due to the fact that it takes, at least, two months to make the mouse model and 45 days to finish the microarray by the company, we worried that we could not finish this microarray analysis if we re-make the mice model with which to perform microarray within 3 months limited by the journal. Therefore, we used the whole livers that have been used for the experiments of *in vivo* knockdown of lnc-LFAR1 to perform microarrays of lnc-LFAR1 on extracellular matrix gene expression. The data are shown in Fig. 4a-e and Supplementary Fig. 9c, d.

The microarray data revealed that 1598 mRNAs were up-regulated and 620 mRNAs were down-regulated in the CCl₄-treated mice infected with lenti-NC, while only 292 mRNAs were up-regulated and 112 mRNAs were down-regulated in the CCl₄-treated mice infected with lenti-shLFAR1. Moreover, there are 133 mRNAs were up-regulated and 178 mRNAs were down-regulated in the CCl₄-treated mice infected with lenti-shLFAR1, compared with the CCl₄-treated mice infected with lenti-NC (Fig. 4b). Among these, 234 out of 311 genes were in common with the set of genes deregulated in the CCl₄-treated mice infected with lenti-NC (Fig. 4c). Additionally, the GO and KEGG pathway analysis revealed that lnc-LFAR1 silencing affects a list of genes associated with collagen fibril organization and TGF β receptor signaling pathway (Fig. 4d,e and Supplementary Fig. 9c,d).

4. Will fully activated HSCs with high expression of lnc-LFAR1 at day 14 still exhibit knockdown of post-fibrogenic genes with lnc-LFAR1-shRNA? Moreover, will there still be minimal TGF β 1 stimulation of post-fibrogenic genes when lnc-LFAR1 is knocked down at later stage of HSCs? Once the fibrotic program gets activated, can it still be reversed, downregulated through regulation of lnc-LFAR1?

Answer 4: Thank you for your nice suggestion. We have knocked down lnc-LFAR1 in activated HSCs

at day 12 and performed qPCR to determine the effect of lnc-LFAR1 knockdown on post-fibrogenic genes. The results showed that it still exhibit knockdown of post-fibrogenic genes with lnc-LFAR1-shRNA, as shown in Fig. 3. Moreover, we have treated HSCs with TGFβ1 when lnc-LFAR1 is knocked down at later stage and found that it will still be minimal TGFβ1 stimulation of post-fibrogenic genes, suggesting once the fibrotic program gets activated, it is still down-regulated through regulation of lnc-LFAR1. The results are shown in Fig. 3a, b.

5. Supplementary Figure 5-8 illustrate that knockdown of lnc-LFAR1 reduces TGFβ-induced pro-apoptosis gene expression and apoptosis in HCs, suggesting that low levels of lnc-LFAR1 in HCs is crucial for deterring HSCs activation. Though, lnc-LFAR1 is already downregulated in primary HCs isolated from mice treated with CCL4 according to Supplementary Figure 4a. How do the results described in Supplementary Figure 5-8 explain the phenomenon in Supplementary Figure 4?

Answer 5: Thank you for your nice suggestion. Liver fibrosis is a result of the wound-healing response of the liver to repeated injury. It is known after an acute liver injury parenchymal cells regenerate and replace the necrotic or apoptotic cells. If the hepatic injury persists, then eventually the parenchymal cells regeneration fails, leading to increased HCs apoptosis, HSCs activation and liver fibrosis.

In this study, we found that lnc-LFAR1 is up-regulated in primary HCs and AML12 cells upon TGFβ treatment, and knockdown of lnc-LFAR1 reduces TGFβ-induced pro-fibrogenic gene expression and HCs apoptosis (Supplementary Figure 5-8), suggesting that lnc-LFAR1 promotes apoptosis of HCs.

We investigated the level of lnc-LFAR1 in CCL₄- and BDL- induced fibrotic livers at different time points. Interestingly, we found the level of lnc-LFAR1 is decreased drastically at 2 weeks after CCL₄ injection and 3 days after BDL (both are early stage of liver fibrosis), while it is gradually increased with persist injury (Supplementary Figure 4d-f). As mentioned above, at early stage of liver fibrosis parenchymal cells regenerate and replace the necrotic or apoptotic cells, and at the same stage lnc-LFAR1 is down-regulated, thus apoptosis of HCs is decreased. The primary HCs used in Supplementary Figure 4a were isolated at 6 weeks after CCL₄ injection and the results showed that lnc-LFAR1 is decreased. At this stage, although lnc-LFAR1 is decreased, compared with normal control, it is increased compared with that at 2 weeks (Supplementary Figure 4f), indicating the

apoptosis of HCs is increased. This could explain the finding that lnc-LFAR1 promotes apoptosis of HCs (Supplementary Figure 5-8), but its level is reduced, compared with normal control (Supplementary Figure 4a), at the stage of 6 weeks after CCl₄ injection.

We have added the explanation of this question in the results and discussion.

6. The paper presents the idea that TGFb induces lnc-LFAR1 expression through Smad2/3 binding at the promoter region of lnc-LFAR1, which in turn positively feeds back into TGFb activation via Smad2/3 phosphorylation and thus progression of liver fibrosis. Figure 6 illustrates upregulation of pSmad2/3 with increased lnc-LFAR1 expression, and decreased pSmad2/3 with lnc-LFAR1 knockdown. Please show and explain how lnc-LFAR1 is having an effect on both total Smad2/3 levels as well as pSmad2/3 independent of total Smad2/3 levels. Data here is required. Otherwise, we could conclude that lnc-LFAR1 has an indirect role rather than represent a direct functional mechanism.

Answer 6: Thank you for your nice suggestion. We have performed experiments to investigate the effect of lnc-LFAR1 on the expression of Smad2 and Smad3, and data showed that lnc-LFAR1 interacts with Smad2/3 (Fig. 8a) to increase the binding of Smad2/3 to the promoters of Smad2 and Smad3 (Fig. 8c) to promote the expression of both genes and thus increasing the amount of total Smad2/3 levels (Fig. 6a-c).

In addition, we also performed experiments to investigate how lnc-LFAR1 affects Smad2/3 phosphorylation. As shown in Fig. 6 (e-h) and Supplementary Fig. 13, lnc-LFAR1 promotes the association of Smad2/3 with TGFβR1 which subsequently phosphorylates Smad2/3.

Taken together, these data revealed the different mechanisms of lnc-LFAR1 in regulating total Smad2/3 levels and phosphorylation of Smad2/3 according to its cellular localization.

Minor Errors and Clarifications:

1. Please clarify how the clusters were defined in the microarray in Figure 1d and Supplementary Figure 1b, and how the clusters were compared between normal and fibrotic livers. The manuscript states that 266 lncRNAs and 1007 mRNAs are upregulated, and 447 lncRNAs and 519 mRNAs are downregulated in the fibrotic livers. From our interpretation of the microarray, there are more lncRNAs

that are upregulated than downregulated and more mRNAs that are downregulated than upregulated in CCL4 mice, which is the opposite trend of what is written.

Answer 1: Thank you for your nice suggestion. We are so sorry for this mistake for turning over the group label on heat map. Thank you for pointing it out.

2. Given that lnc-LFAR1 is present on the list of 10 randomly selected lncRNAs from hundreds of differentially regulated lncRNAs found in fibrotic liver, the selection process to validate microarray analysis does not seem random. Please clarify how these 10 lncRNAs are selected.

Answer 2: Thank you for your nice suggestion. Ten lncRNAs were chosen according to the fold change and the lncRNA-mRNA co-expression network. We have corrected this in the manuscript.

3. Please show relative level of mRNA for pro-fibrotic, pro-inflammation, and pro-apoptotic genes normalized to GAPDH in primary HSCs and primary HCs.

Answer 3: Thank you for your nice suggestions. We have shown the relative level of mRNA for these genes and indicated that they are normalized to GAPDH in the figure legends, as shown in Fig. 3, Supplementary Fig. 6 and Supplementary Fig. 8.

4. Were the C57 mice also randomly divided into 4 groups and received the same course of treatment as Balb/c mice?

Answer 4: Thank you for your nice suggestions. C57 mice were randomly divided into 2 groups, as normal control and CCl₄-induced liver fibrosis model, to verify results of lncRNA microarray analysis.

5. Only 7 representative mRNAs are shown in Supplementary Figure 1c and 9 in Supplementary Figure 1d.

Answer 5: Thank you for your nice suggestions. We have performed experiments with the other genes and added the results in Supplementary Fig. 1c and 1d.

6. Is there significant different of lnc-LFAR1 between HSCs and Kupffer cells in Figure 2b?

Answer 6: Thank you for your nice suggestion. There is no significant difference of lnc-LFAR1 between HSCs and Kupffer cells.

7. Are the primary HSCs cultured at day 12 culture-activated prior to TGF β 1 stimulation and exhibit increase in level of α SMA and lnc-LFAR1 as shown in Figure 2d? If so, then is the increase in relative level of mRNA from TGF β 1 stimulation a fold increase on top of culture-induced activation in Figure 2f?

Answer 7: Thank you for your questions. Yes, the primary HSCs cultured at day 12 are culture-activated prior to TGF β 1 stimulation and exhibit increase in level of α -SMA and lnc-LFAR1, as shown in Figure 2d.

Yes, the increase in relative level of mRNA from TGF β 1 stimulation is a fold increase on top of culture-induced activation, as shown in Figure 2f.

8. Why is the level of TGF β 1 lower in both LV-control and LV-lnc-LFAR1 in comparison to that in shRNA-control in Figure 3?

Answer 8: Thank you for your nice suggestions. This was caused by different exposure time and we have refined western blot data in Fig. 3e, from which endogenous TGF β 1 can be shown.

9. Misspelled “binding” in line 263.

Answer 9: Thank you for your nice suggestions. We have carefully checked the spelling and corrected the mistakes in the manuscript.

Summary: The linkage of this lncRNA, LFAR1, to inflammatory genes and the fibrotic program is potentially of general interest and significance, but a more robust data set for mechanistic insight is required. Absent this, the paper falls short of the level of Nature Communications.

Answer: Thank you for your nice suggestions. We have performed experiments to investigate the effect of lnc-LFAR1 on the expression of Smad2 and Smad3, and data showed that lnc-LFAR1 interacts with Smad2/3 (Fig. 8a) to increase the binding of Smad2/3 to the promoters of Smad2 and Smad3 (Fig. 8c) to promote the expression of both genes and thus increasing the amount of total Smad2/3 levels (Fig. 6a-c). In addition, we also performed experiments to investigate how lnc-LFAR1 affects Smad2/3 phosphorylation. As shown in Fig. 6 (e-h) and Supplementary Fig. 13, lnc-LFAR1 promotes the association of Smad2/3 with TGF β R1 which subsequently phosphorylates Smad2/3. Taken together,

these data revealed the different mechanisms of lnc-LFAR1 in regulating total Smad2/3 levels and phosphorylation of Smad2/3 according to its cellular localization.

Also, we have made a BDL-induced fibrosis mouse model to investigate the role of lnc-LFAR1. We hope all the data for mechanistic insight can reach the level of journal. Thanks again for your helpful comments.

Reviewers' comments:

Reviewer #1 (Remarks to the Author):

The authors have made extensive, constructive efforts to address the prior concerns. One lingering issue that still needs clarification is the role and activity of lnc-LFAR1 in hepatocytes. Despite this and the authors' prior publications, this cell type is NOT a major source of fibrogenic gene expression and TGFbeta responsiveness during fibrosis in vivo. Thus, the studies claiming this are at odds with an extensive published literature. Along these lines, the findings do not sufficiently clarify the relative importance of lnc-LFAR1 in hepatocytes vs. stellate cells. Is antagonism of lnc-LFAR1 leading to reduced injury from hepatocytes, (which is suggested by a 50% reduction in AST and ALT) or antagonism of stellate cell fibrogenesis? Also supporting this concern is the alpha SMA staining which is not at all localized to stellate cells as it should be (Figure 1), but rather is apparent in hepatocytes.

Reviewer #2 (Remarks to the Author):

Zhang et al. clearly address an intriguing idea about the role of a tissue-restricted lncRNA on liver biology. In fact, simply because the finding is so striking, it is necessary to exercise a bit more caution with respect to the supporting evidence. Hence even though the authors were able to address and clarify majority of our points of concern. Unfortunately, the paper still lacks sufficiently convincing evidence for mechanistic insight in terms of how lnc-LFAR1 regulates the expression of pro-fibrogenic, pro-inflammatory, and apoptotic-related genes, given the scope of the claims. We do note that the presented data do seem consistent, and if valid, suggest a novel mechanism in driving liver fibrogenesis via a lnc-RNA. Therefore, we would urge the following additional experiments:

Major Points:

1. Please show additional controls for the knockdown data. It is understandable that only sh-lnc-LFAR1-1 and sh-lnc-LFAR1-3 were presented as they showed the most efficient knockdown. However, it is suspicious that both knockdowns of a single factor, especially a lnc-RNA, is capable of dramatically depleting expression of all extracellular matrix genes of interest in the liver. Thus, one should really perform an RNA-seq to give a clear answer- and it does not matter if this takes a few extra weeks- it makes the conclusions more rigorous.
2. The paper proposes two different mechanisms in terms of how lnc-LFAR1 regulates Smad2/3 and pSmad2/3.
 - a. Though RIP qPCR shows a significant association of lnc-LFAR1 with Smad2/3 and ChIP qPCR shows increased binding of Smad2/3 at promoter regions of Smad2 and Smad3 upon overexpressing lnc-LFAR1, it is difficult to conclude the specificity of these associations without deep sequencing. Please provide this.
 - b. Similarly, RIP-seq will confirm the specificity of lnc-LFAR1-TGFBR1 interaction mediating the increased phosphorylation of Smad2/3.
3. With these data available to inspect, the paper would pass to the level of clearly publishable.

Response to Reviewers

Reviewers' comments:

Reviewer #1 (Remarks to the Author):

The authors have made extensive, constructive efforts to address the prior concerns. One lingering issue that still needs clarification is the role and activity of lnc-LFAR1 in hepatocytes. Despite this and the authors' prior publications, this cell type is NOT a major source of fibrogenic gene expression and TGFbeta responsiveness during fibrosis *in vivo*. Thus, the studies claiming this are at odds with an extensive published literature. Along these lines, the findings do not sufficiently clarify the relative importance of lnc-LFAR1 in hepatocytes vs. stellate cells. Is antagonism of lnc-LFAR1 leading to reduced injury from hepatocytes, (which is suggested by a 50% reduction in AST and ALT) or antagonism of stellate cell fibrogenesis? Also supporting this concern is the alpha SMA staining which is not at all localized to stellate cells as it should be (Figure 1), but rather is apparent in hepatocytes.

Answer: Thank you for your nice suggestion. In this study, we demonstrated that knockdown of lnc-LFAR1 in primary HSCs isolated from normal mice dramatically decreased TGFβ-induced up-regulation of the pro-fibrogenic genes (Fig. 3f-h), and antagonism of lnc-LFAR1 in primary HSCs isolated from lenti-shLFAR1-infected mice, in comparison with the lenti-NC mice, decreased CCl₄- and BDL-induced up-regulation of these pro-fibrogenic genes (Supplementary Fig. 9f and Supplementary Fig. 11b). For the HCs, we demonstrated that knockdown of lnc-LFAR1 reduces TGFβ-, CCl₄- and BDL-induced pro-fibrogenic, pro-inflammation and pro-apoptosis genes expression and apoptosis of HCs (Supplementary Fig. 5, 6, 8; Supplementary Fig. 9e, h, i and Supplementary Fig. 11c, d). Moreover, antagonism of lnc-LFAR1 leading to reduced injury from hepatocytes, which is suggested by the reduction in AST and ALT (Supplementary Table 1 and 2). Taken together, our data revealed that antagonism of lnc-LFAR1 attenuates liver fibrosis *in vitro* and *in vivo* through inhibiting the activation of HSCs and repressing the apoptosis of HCs. We have refined the results and discussion of the manuscript to shed light on the importance of lnc-LFAR1 in hepatocytes and HSCs. In addition, we have refined α-SMA staining in Fig. 1a.

Once again, thanks for your nice comments. We deeply appreciate the time and effort you've spent in our manuscript.

Reviewer #2 (Remarks to the Author):

Zhang et al. clearly address an intriguing idea about the role of a tissue-restricted lncRNA on liver biology. In fact, simply because the finding is so striking, it is necessary to exercise a bit more caution with respect to the supporting evidence. Hence even though the authors were able to address and clarify majority of our points of concern. Unfortunately, the paper still lacks sufficiently convincing evidence for mechanistic insight in terms of how lnc-LFAR1 regulates the expression of pro-fibrogenic, pro-inflammatory, and apoptotic-related genes, given the scope of the claims. We do note that the presented data do seem consistent, and if valid, suggest a novel mechanism in driving liver fibrogenesis via a lnc-RNA. Therefore, we would urge the following additional experiments:

Major Points:

1. Please show additional controls for the knockdown data. It is understandable that only sh-lnc-LFAR1-1 and sh-lnc-LFAR1-3 were presented as they showed the most efficient knockdown. However, it is suspicious that both knockdowns of a single factor, especially a lnc-RNA, is capable of dramatically depleting expression of all extracellular matrix genes of interest in the liver. Thus, one should really perform an RNA-seq to give a clear answer- and it does not matter if this takes a few extra weeks- it makes the conclusions more rigorous.

Answer 1: Thank you for your nice suggestion. We have knocked down lnc-LFAR1 with sh-lnc-LFAR1-1 and sh-lnc-LFAR1-3 in primary HSCs and isolated RNA for RNA-seq. The data revealed that 1195 mRNAs were up-regulated and 1424 mRNAs were down-regulated in the primary HSCs infected with lncRNA-shRNA-1. Among these, 2023 out of 2619 genes, including *α-SMA*, *Coll1a1*, *Coll1a2*, *Col3a1*, *Col4a5*, *TGFβRI*, *MMPs*, and *TIMPs*, were in common with the set of genes deregulated in the primary HSCs infected with lncRNA-shRNA-3 (Fig. 3a-c). The GO analysis revealed that lnc-LFAR1 silencing affects a list of genes associated with extracellular matrix and the KEGG pathway analysis demonstrated ECM-receptor interaction pathway (Fig. 3d, e), suggesting that lnc-LFAR1 regulates the expression of extracellular matrix genes in HSCs. These data were consistent with the results of qRT-PCR, which makes the conclusions more rigorous. The transcriptome

sequencing data have been deposited in NCBI Gene Expression Omnibus (GEO) under the following accession number: GSE96526.

2. The paper proposes two different mechanisms in terms of how lnc-LFAR1 regulates Smad2/3 and pSmad2/3.

a. Though RIP qPCR shows a significant association of lnc-LFAR1 with Smad2/3 and ChIP qPCR shows increased binding of Smad2/3 at promoter regions of Smad2 and Smad3 upon overexpressing lnc-LFAR1, it is difficult to conclude the specificity of these associations without deep sequencing. Please provide this.

Answer 2a: Thank you for your nice suggestion. Over the past months, we have ChIPed the promoter regions with Smad2/3 antibody by three independent experiments in primary HSCs isolated from 40-week aged mice for ChIP-seq. However, the samples of the DNA from these three experiments were not qualified for sequencing (the company informed us that the ChIPed DNA did not pass the quality control), which needs large amount of DNA and fragments between 100 – 500 bp. It seems like not feasible to perform the ChIP-seq because we could not collect large amount of primary HSCs and the 40-week aged mice are not enough within the limited period. Therefore, we reflected this difficulty to the editor and asked whether we can perform additional experiments and provide other evidences to answer this question and was allowed to provide these data. Please evaluate the following data whether these could answer the question and we look forward to hearing the comments and accept the criticism.

Firstly, to confirm the specificity, we have also performed ChIP with AML12 cells, and obtained the same results as shown with primary HSCs (Fig. 8c). We have added this result in the revised manuscript (Supplementary Fig. 17c).

Secondly, in our previous revised manuscript, we have shown the positive control PAI and negative control GAPDH in ChIP experiment to confirm the binding specificity (Fig. 8c).

Thirdly, we have actually applied three pairs of primers for each gene promoter and our data reveal increased binding of Smad2/3 at promoter regions of Smad2 and Smad3 upon overexpressing

lnc-LFAR1. This comes from the qRT-PCR results with primers amplifying Smad2 (-1076 - -1002), Smad2 (-1853 - -1733) and Smad3 (-1271 - -1038) (Supplementary Table 3) which were consistent with the prediction of Smad2/3 binding site at promoter regions of Smad2 and Smad3 by JASPAR and PROMO. However, application of other sets of primers amplifying Smad2 (-222 - -80), Smad3 (-198 - -7) and Smad3 (-1952 - -1778) (Supplementary Table 3) showed no binding of Smad2/3 at promoter regions of Smad2 and Smad3 genes, which can act as a negative control to confirm the specificity of these associations. We could not present all the results with three sets of primers for eleven target genes, but only showed the final results and listed all the primers in Supplementary Table 3.

Fourthly, ChIP-qPCR showed no increased binding of Smad2/3 at the promoters of other target genes including TIMP1 and Hes1 after overexpression of lnc-LFAR1 (Fig. 8c), further confirming the specificity of binding of Smad2/3 at promoter regions of Smad2 and Smad3.

Taken together, we hope the above additional experiments and evidences could confirm the specificity of the associations and hope these can be accepted.

b. Similarly, RIP-seq will confirm the specificity of lnc-LFAR1-TGFBR1 interaction mediating the increased phosphorylation of Smad2/3.

Answer 2b: Thank you for this suggestion. Actually, to confirm the specificity of lnc-LFAR1-TGFBR1 interaction, we have applied three sets of primers (Supplementary Table 3) to amplify lnc-LFAR1 after RIP by the TGFBR1 antibody, and all the primers could obtain positive results. In this experiment, we have also used primers to amplify Actin which showed no product could be detected. In our previous version, we did not mention the details but only stating the final result (Fig. 6g). After reading this point of comment, over the past weeks, we have used primers to amplify another lncRNA ENSMUST00000154817 (Supplementary Table 3), identified by the same microarray and shown down-regulation in liver fibrosis, with the same material used to amplify lnc-LFAR1. The result of qRT-PCR showed no amplified product, indicating there is no interaction of ENSMUST00000154817 with TGFBR1. We think this result could be used as a control to eliminate the possibility of false positive interaction of lncRNAs with TGFBR1. We have added these additional data in the revised paper (Fig. 6g) and hope to be accepted that these results could confirm the specificity.

3. With these data available to inspect, the paper would pass to the level of clearly publishable.

Answer 3: We are very grateful for your help, for instance, your comment that requires to explain how lnc-LFAR1 is having an effect on pSmad2/3 helped us identifying TGF β R1 is an mediator, which improves the quality of the manuscript. Although we have tried our best to answer the questions, we know that some of which may be not satisfied. Thanks again for your helpful comments.

REVIEWERS' COMMENTS:

Reviewer #2 (Remarks to the Author):

In the revised manuscript, Zhang et al. have provided additional data through RNA-seq and showed broad consistencies, making their conclusions more rigorous. However, they have failed to provide genome-wide data through ChIP-seq, which is crucial to show the consistency and specificity of knockdown of lnc-LFAR1 with the binding of Smad2/3, although this is shown by ChIP of specific sites. We would think it important to provide the overlap of genes affected by RNA-seq with the genes that are bound from the ChIP-seq data. We understand that Zhang et al. were unable to perform ChIP-seq due to lack of DNA isolated from 40-week old mice. However, they performed ChIP with AML12 cells and obtained similar results compared to that of primary HSCs. This indicates that AML12 cells can be a model cell line from which enough DNA can be harvested for ChIP-seq. The use of AML12 cells can similarly be applied to confirm the specificity of lnc-LFAR1-TGFBR1 interaction in an unbiased manner through RIP-seq. Given the scope of the claims of how lnc-LFAR1 is capable of mediating liver fibrogenesis through two separate mechanisms, deep sequencing is essential to conclude specificity in an unbiased manner. We appreciate the experimental burden imposed by our request for these additional data, but it is required for a rigorous conclusion on the role of a lncRNA in an important biological system.

Response to Reviewers

Reviewers' comments:

Reviewer #2 (Remarks to the Author):

In the revised manuscript, Zhang et al. have provided additional data through RNA-seq and showed

broad consistencies, making their conclusions more rigorous. However, they have failed to provide genome-wide data through ChIP-seq, which is crucial to show the consistency and specificity of knockdown of lnc-LFAR1 with the binding of Smad2/3, although this is shown by ChIP of specific sites. We would think it important to provide the overlap of genes affected by RNA-seq with the genes that are bound from the ChIP-seq data. We understand that Zhang et al. were unable to perform ChIP-seq due to lack of DNA isolated from 40-week old mice. However, they performed ChIP with AML12 cells and obtained similar results compared to that of primary HSCs. This indicates that AML12 cells can be a model cell line from which enough DNA can be harvested for ChIP-seq. The use of AML12 cells can similarly be applied to confirm the specificity of lnc-LFAR1-TGFBR1 interaction in an unbiased manner through RIP-seq. Given the scope of the claims of how lnc-LFAR1 is capable of mediating liver fibrogenesis through two separate mechanisms, deep sequencing is essential to conclude specificity in an unbiased manner. We appreciate the experimental burden imposed by our request for these additional data, but it is required for a rigorous conclusion on the role of a lncRNA in an important biological system.

Answer: Thank you for your nice suggestion. We acknowledge that although we have revealed that lnc-LFAR1 interacts with TGF β R1, by RIP, which subsequently phosphorylates Smad2/3 to promote its nuclear translocation and the binding to the target promoters as shown by ChIP, RIP-seq can further confirm the specificity of lnc-LFAR1-TGF β R1 interaction and ChIP-seq can further confirm the binding of Smad2/3 to specific sites. We acknowledge lack of RIP-seq and ChIP-seq was a limitation of this manuscript, therefore we discuss this point in the discussion. Thank you again.